# An atlas of the binding specificities of transcription factors in *Pseudomonas aeruginosa* directs prediction of novel regulators in virulence

Tingting Wang[1†], Wenju Sun[2†], Ligang Fan[1,2†], Canfeng Hua[1†], Nan Wu[2], Shaorong Fan[2], Jilin Zhang[3], Xin Deng[1]*, Jian Yan[1,2]*

[1]Department of Biomedical Sciences, City University of Hong Kong, Kowloon Tong, Hong Kong SAR, China; [2]School of Medicine, Northwest University, Xi'an, China; [3]Department of Medical Biochemistry and Biophysics, Karolinska Institutet, Solna, Sweden

*For correspondence:
xindeng@cityu.edu.hk (XD);
jian.yan@cityu.edu.hk (JY)

[†]These authors contributed equally to this work

Competing interests: The authors declare that no competing interests exist.

**Abstract** A high-throughput systematic evolution of ligands by exponential enrichment assay was applied to 371 putative TFs in *Pseudomonas aeruginosa*, which resulted in the robust enrichment of 199 unique sequence motifs describing the binding specificities of 182 TFs. By scanning the genome, we predicted in total 33,709 significant interactions between TFs and their target loci, which were more than 11-fold enriched in the intergenic regions but depleted in the gene body regions. To further explore and delineate the physiological and pathogenic roles of TFs in *P. aeruginosa*, we constructed regulatory networks for nine major virulence-associated pathways and found that 51 TFs were potentially significantly associated with these virulence pathways, 32 of which had not been characterized before, and some were even involved in multiple pathways. These results will significantly facilitate future studies on transcriptional regulation in *P. aeruginosa* and other relevant pathogens, and accelerate to discover effective treatment and prevention strategies for the associated infectious diseases.

## Introduction

*Pseudomonas aeruginosa* is an opportunistic human pathogen of considerable medical importance, as it causes pneumonia that is associated with high morbidity and mortality rates in immunocompromised patients, burn victims, and cystic fibrosis patients (*Williams et al., 2010*; *Honda et al., 1977*). Annually, more than 2 million patients are infected by the pathogen, and approximately 90,000 people die from it (*Cross et al., 1983*). The regulation of virulence-related pathways is mainly under the control of a large group of sequence-specific transcription factors (TFs) (*Papavassiliou and Papavassiliou, 2016*; *Huang et al., 2019*). TFs orchestrate the transcription of downstream genes and guide the expression of the genome by recognizing and occupying target promoter regions using their DNA-binding domains, thereby supporting or blocking the recruitment of RNA polymerase (*Lambert et al., 2018*; *Wade, 2015*). In eukaryotes, for example, the aberrant activity of human TFs or mutation in TF-binding sites (TFBSs) causes diseases such as cancers, cardiovascular disease, diabetes, obesity, and inflammation (*Papavassiliou and Papavassiliou, 2016*). The TFBSs on genomic DNA are predominantly determined by the sequence specificities of TFs, a fundamental property with which to unravel the functions of TFs and their roles in the mechanisms underlying disease etiology.

Although nearly 400 transcriptional regulators were previously predicted, only approximately 30 regulators have been characterized as virulence-associated TFs in *P. aeruginosa* over the past

decades (*Huang et al., 2019*). To date, less than 5% of the *P. aeruginosa* TFs have been profiled for DNA-binding specificities, and thus, the downstream genes or upstream regulators of TFs remain largely unknown. To fill this gap, we applied a high-throughput systematic evolution of ligands by exponential enrichment (HT-SELEX) assay (*Jolma et al., 2013*) to all putative TFs (371 TFs) in the *P. aeruginosa* PAO1 genome, and successfully obtained 199 position weight matrix (PWM) models describing the DNA-binding specificities of 182 TFs. With the obtained PWM models, we scanned the PAO1 genome and reported 33,709 putative TFBSs that were highly enriched in the intergenic regions encompassing regulatory elements compared with gene body regions (odds ratio = 11.75). TF–target interactions were established for nine virulence-associated pathways, which enabled global decoding of the pathogenic regulatory networks of this microorganism. We validated some binding sites through a series of biochemical and genetic experiments and, interestingly, identified novel TFs potentially implicated in pathogenesis. This study provides an unprecedented scale of high-quality data on TF-binding specificity in a single bacterium and a global view of transcriptional regulatory relationships in *P. aeruginosa*, which constitute a major step toward deciphering the regulatory mechanisms of virulence. The findings are expected to substantially facilitate the development of effective therapies for the associated infectious disease.

## Results

### HT-SELEX reveals binding specificities of 182 TFs in *P. aeruginosa*

According to the existing annotations in 'Pseudomonas Genome DB' (https://www.pseudomonas.com/) (*Winsor et al., 2016*), the *P. aeruginosa* PAO1 genome contains 371 putative TFs, which can be classified into 29 function-associated families (*El-Gebali et al., 2019*; *Pérez-Rueda and Collado-Vides, 2000*). The vast majority of *P. aeruginosa* TFs (269/371) belong to six families: the LysR family, AraC family, LuxR family, OmpR family, TetR family, and GntR family (*Supplementary file 1A–C*). To determine the DNA-binding specificity, we performed a well-established HT-SELEX assay (*Jolma et al., 2013*) for all the 371 putative TFs, including 262 (71%) probable transcriptional regulators, 64 (17%) previously annotated TFs, and 45 (12%) cognate response regulators (RRs) in the two-component systems (TCS) (*King et al., 1954*) with predicted DNA-binding characteristics. Briefly, each full-length protein fused with a C-terminal 6×His-tag was expressed in and purified from *Escherichia coli*. The TFs were then subjected to four rounds of HT-SELEX enrichment, starting with a double-stranded DNA input library consisting of 40 bp randomized sequences adapted to the Illumina or BGI parallel sequencing system (see *Figure 1A*, *Figure 1—figure supplement 1A–D*, *Supplementary file 1D* and Materials and methods for details of the input design and experimental procedure). The sequencing reads from each consecutive HT-SELEX round were analyzed using a previously developed 'multinomial' algorithm (*Nitta et al., 2015*), which finally generated 199 binding profiles (PWM models) for 182 TFs. Some of these TFs were found to bind to DNA in homodimeric modes with different spacing and monomer orientations (*Jolma et al., 2013*), each represented by a PWM model (*Stormo, 2013*; *Figure 1A*, *Supplementary file 2*). Twelve independent replicates with separate protein purification and input DNA synthesis procedures were generated, and each pair of replicates showed virtually identical binding specificities, thus demonstrating the high reproducibility of HT-SELEX findings (highlighted in *Supplementary file 2*). In the subsequent analyses, we only included 199 unique PWM models by retaining one PWM model from each pair of technical replicates. Of the top six largest TF families, the OmpR, GntR, and LysR families demonstrated motif enrichment for more than 50% of their TF members (71%, 61%, and 52%, respectively; *Supplementary file 1C*). Taken together, HT-SELEX generated TF-binding motifs from 23 different families, with an overall success rate of 49% (182/371) for the entire TF repertoire of *P. aeruginosa* (*Supplementary file 1B,C*).

### TF classification by DNA-binding specificity

The DNA-binding specificity of a TF determines where it binds to the genome and which genes it regulates, and this property has notable effects on the physiological activity of the organism. We next generated TF regulatory networks to compare their sequence preferences and grouped the 182 TFs according to the similarity of their DNA-binding specificity. The similarity analysis using SSTAT (*Jolma et al., 2013*; *Pape et al., 2008*) (see Materials and methods) revealed that 133 PWM

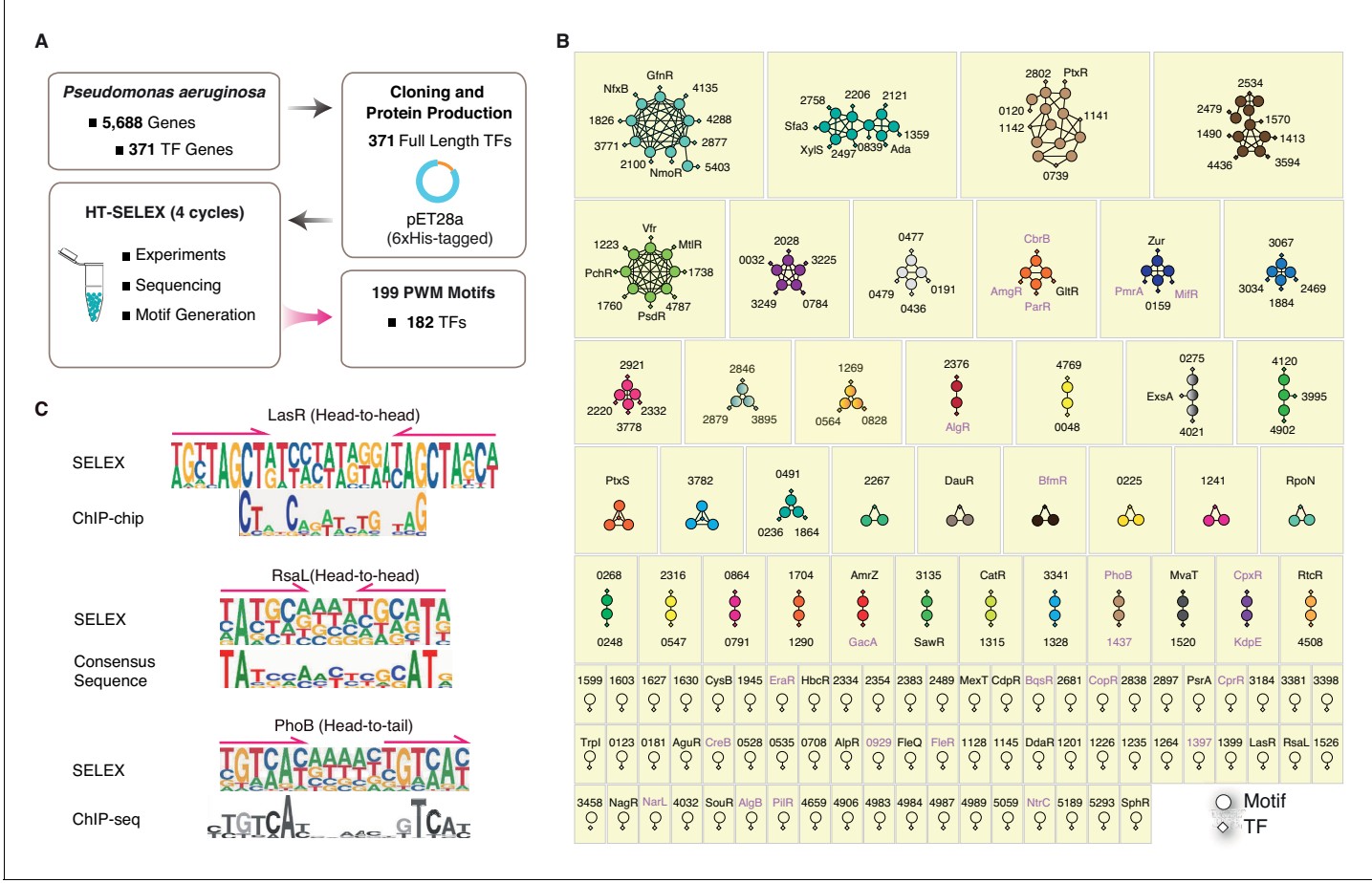

**Figure 1.** Summary of the HT-SELEX results in *P. aeruginosa*. (**A**) Schematic description of protein expression and HT-SELEX procedure and output. (**B**) Network analysis of similarity of the obtained PWMs. Diamonds indicate TF genes, circles indicate individual PWMs. Edges are drawn between a TF and its PWM model, and between similar models if SSTAT similarity score >1.5e-05. To save space, 'PA' is omitted in all TFs names. The names of RRs of TCS are marked in pink font. (**C**) Comparison of the binding motifs of three TFs (LasR, RsaL, and PhoB) obtained from HT-SELEX (upper) and ChIP or cross-species sequence alignment methods (lower). Arrows indicate half-sites in dimeric sites. Also see *Figure 1—figure supplement 1*, *Supplementary file 1*.

The online version of this article includes the following figure supplement(s) for figure 1:

**Figure supplement 1.** Ligand design of HT-SELEX and the distribution of the 24 RRs genomic binding, related to *Figure 1A* and *Figure 1B*, and *Figure 2D*.

models from 116 TFs formed 38 interconnected subnetworks, each containing at least two TFs (*Figure 1B*). In addition, we observed 66 isolated subnetworks without significant connections to any other TF, which illustrated the generally diverse DNA-binding specificity of most TFs in *P. aeruginosa* (*Figure 1B*). In summary, the network analysis categorized 104 clusters of different DNA sequence preferences for 182 TFs in *P. aeruginosa* (*Supplementary file 3*).

## Most TFs bind to DNA sequences in a homodimeric manner

A comparison between motifs previously established using different methods, e.g. chromatin immunoprecipitation (ChIP) (*Gilbert et al., 2009*; *Shao et al., 2018*) and cross-species consensus sequence alignment (*Kang et al., 2017*), and those identified in the present study revealed identical or similar motifs for LasR (*Gilbert et al., 2009*), RasL (*Kang et al., 2017*), and PhoB (*Huang et al., 2019*; *Bielecki et al., 2015*), demonstrating the high quality of PWM models obtained from our HT-SELEX analysis (*Figure 1C*). In general, HT-SELEX-generated motifs were also longer than the in vivo motifs generated from ChIP data, most likely because these in vivo motifs, compared with HT-SELEX-generated motifs, were derived from a much smaller number of binding sites owing to the

small genome size. We could clearly identify two tandem homodimeric half sites (TAGCT) in the HT-SELEX-derived motif of LasR, whereas its ChIP-derived motif turned out sparse and contained only part of the monomeric site (CT), indicating the potentially poor quality of the previously known in vivo motifs.

Analysis of the PWM model length revealed that the motifs ranged from 9 to 28 bp in *P. aeruginosa*, with the most prevalent length being 16 bp (*Figure 2A*, *Supplementary file 2*). These long binding sites indicate that many TFs tend to bind to DNA sequences in a homodimeric manner. Indeed, the vast majority (179/199) of the PWM models displayed homodimeric-type binding (*Figure 2B*, *Supplementary file 2*), whereas only approx. 10% (20/199) of the PWM models exhibited monomeric binding specificity. Among the 20 monomeric PWM models, 16 TFs had only one PWM model while some TFs, such as PA1241, yielded two monomeric PWM models with different spacing between the two half sites (*Figure 1B*, *Supplementary file 3*). In addition, half of the TFs with monomeric motifs (nine TFs) belonged to the LysR and AraC families, the two largest families of TFs in *P. aeruginosa* PAO1. The monomeric sites were generally shorter than the dimeric sites and were therefore more prevalent in the genome, allowing higher flexibility in the transcriptional regulation of a broad range of genes. This suggests that these monomeric TFs play multiple regulatory roles. Most TFs exhibited two identical protein molecules bound in opposite orientations on different DNA strands forming a head-to-head homodimer, whereas only 16 TFs showed a consecutive binding orientation in the same direction (head-to-tail) (*Figure 2B*). Early studies have shown that some TF dimer formation events depend on the DNA molecules they interact with, and display strong enrichment for specific monomer orientation and spacing, unlike independent binding events by two monomers that may allow any spacing or orientation between them (*Jolma et al., 2013*; *Jolma et al., 2015*). For example, the TF PhoB binds to DNA as a homodimer in a head-to-tail consecutive orientation, with a 'GTCA(C/T)' monomer sequence preference spaced by a stretch of 6 bp AT-rich nucleotides (*Figure 1C*), supported by independent sets of ChIP-seq experiments (*Huang et al., 2019*; *Bielecki et al., 2015*). Consistently, many TFs have been confirmed to act as dimers when binding to DNA, such as LasR (*Bottomley et al., 2007*; *Fan et al., 2013*), RsaL (*Kang et al., 2017*), PsrA (*Kang et al., 2009*; *Kojic et al., 2002*), FleQ (*Su et al., 2015*), and QscR (*Wysoczynski-Horita et al., 2018*). Interestingly, most of the head-to-tail homodimeric TFs were found to belong to the OmpR family, which is also the main family for the RRs in the TCS (*Figure 1B*, *Supplementary file 1B* and *Supplementary file 2*). The head-to-head orientation is generally preferred by TFs across species, including humans (*Jolma et al., 2015*).

To predict direct interactions between TFs and DNA sequences, we first used the FIMO software (*Grant et al., 2011*) to scan genome-wide TFBSs for all of the 182 TFs with available PWM models in the *P. aeruginosa* PAO1 reference genome and identified 33,709 significant putative TFBSs (p<1.0e-5). For individual TFs, the number of TFBSs differed substantially, with more than 36% (65/182) of TFs predicted to bind to over 100 sites in the 6.3 Mb genome (*Figure 2C*, *Supplementary file 4*). The putative TFBSs were highly enriched in the intergenic regions than in the gene body regions (negative binomial test, p<2.2e-16), with 57% of them densely located within only a 10% fraction of the genome (*Figure 2D*). Some TFs were predicted to bind to only intergenic regions, such as PA4776 and PA5511 (*Figure 1—figure supplement 1E*). To verify the PWM-predicted binding sites, we performed electrophoretic mobility shift assays (EMSAs) using recombinant TF proteins and cloned double-stranded DNA fragments containing the predicted genomic binding sites for the tested TFs. This method was similar to HT-SELEX but avoided exponential competition between DNA sequences. EMSA could be more sensitive than HT-SELEX in detecting some weak binding sites but was limited in throughput in terms of the number of DNA sequences analyzed. In total, EMSA successfully validated 62 pairs of HT-SELEX motif-predicted TF–DNA interactions, including the binding modes (monomeric vs homodimeric binding) of the TFs in each interaction and various monomer spacings and orientations, thus demonstrating the high quality of the data set (*Figure 2—figure supplements 1–11*).

Previous studies have shown that the vast majority of TF binding within the gene body region has little or no effect on the transcriptional level in prokaryotes (*Shimada et al., 2008*); therefore, we primarily focused on TFBSs in the intergenic regions in the subsequent analyses. We were particularly interested in investigating the transcriptional regulatory program in relation to the virulence-associated growth and pathogenesis of *P. aeruginosa*. To colonize and overwhelm host tissues, nine pathways function in *P. aeruginosa* to exert its virulence, including biofilm production (*Whiteley et al.,*

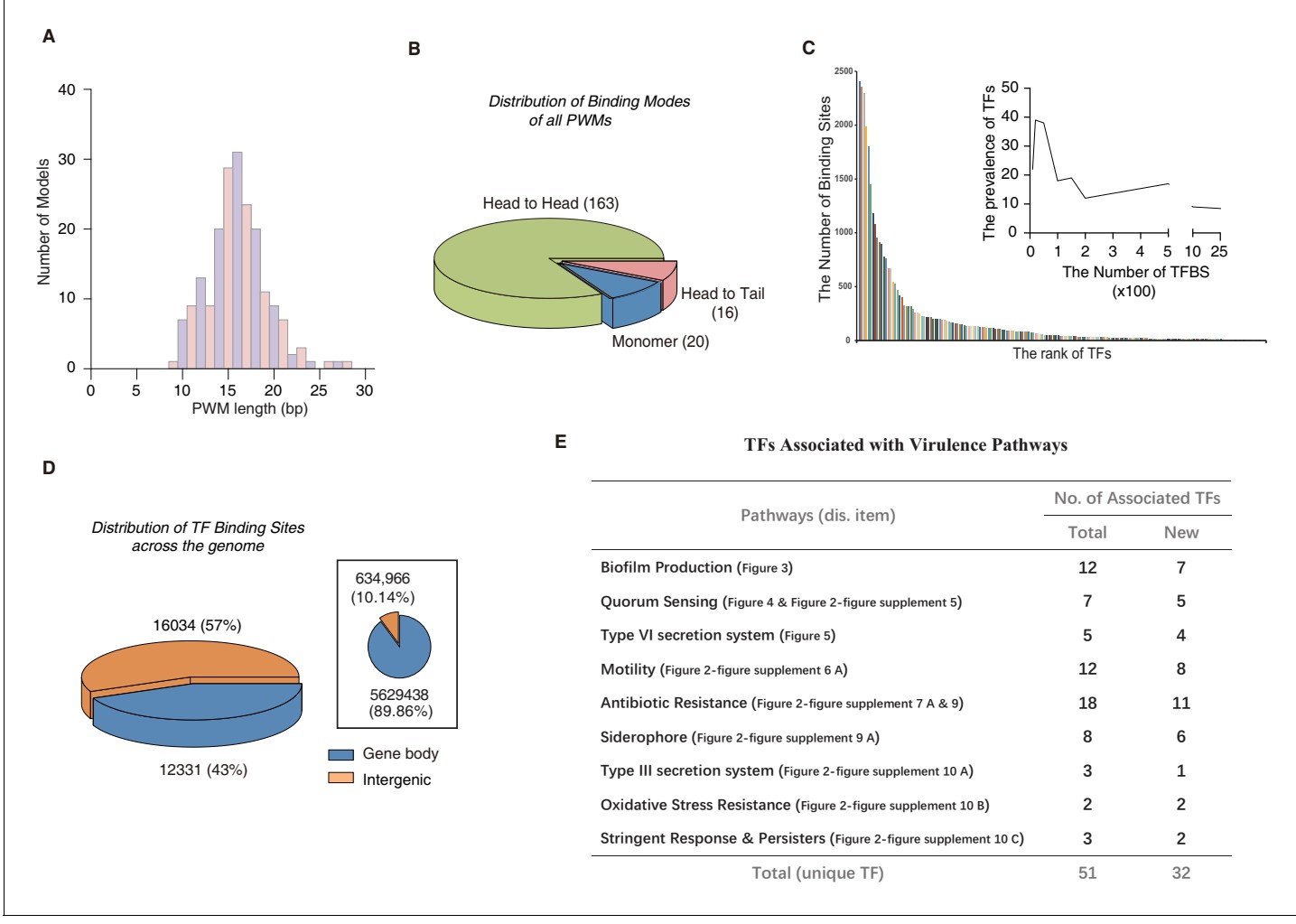

**Figure 2.** Comparison of different TF-binding modes. (**A**) Histogram shows the distribution of the lengths of all PWM models, using red for odds numbers and blue for even numbers for better illustration. (**B**) Pie chart shows the category of different TF-binding modes. Classification of all binding models into non-repetitive sites (monomer) and sites with two similar subsequences (dimer). The dimeric types are further classified as head-to-head (two TF protein molecules bind to opposite orientation on DNA) and head-to-tail (two TF protein molecules bind consecutively on the same orientation on DNA). (**C**) Bar chart shows the number of the binding sites per TF. Note that most of TFs target fewer than 100 genes, while eight TFs exceptionally bind to more than 1000 sites in the genome. The inset histogram shows the prevalence of TFs with the corresponding number of predicted TFBSs. (**D**) The position annotation of binding sites of all 182 TFs in the *P. aeruginosa* genome using pie charts. The pie chart area is proportional to the percentage of predicted binding site location for all TFs, either inside (blue) or outside (orange) gene body regions. The inset shows the fraction of gene body (blue) and intergenic (orange) regions in the genome, reflected by the area of the two colors in the pie chart. (**E**) The number of TFs potentially significantly associated with nine virulence-associated pathways. The corresponding transcriptional regulatory network and validation details for each pathway are indicated in the parenthesis (display item). Note that newly associated TFs indicate that the TFs are uncharacterized genes. Also see *Figure 2—figure supplement 12*.

The online version of this article includes the following source data and figure supplement(s) for figure 2:

**Source data 1.** Source data for *Figure 2A*.
**Source data 2.** Source data for *Figure 2C*.
**Figure supplement 1.** Validation of different modes of TF binding, related to *Figure 2*.
**Figure supplement 2.** Validation of different modes of PhoB binding, related to *Figure 2*.
**Figure supplement 3.** Validation of different modes of CpxR binding, related to *Figure 2*.
**Figure supplement 4.** Transcriptional regulation in biofilm pathway, related to *Figure 2*.
**Figure supplement 4—source data 1.** Source data for *Figure 2—figure supplement 4B*.
**Figure supplement 5.** Transcriptional regulation in QS pathway, related to *Figure 2*.
**Figure supplement 6.** Transcriptional regulation in motility pathway.
**Figure supplement 7.** Transcriptional regulation in antibiotic resistance pathway.
**Figure supplement 8.** Transcriptional regulation in antibiotic resistance pathway, related to *Figure 2—figure supplement 7*.
*Figure 2 continued on next page*

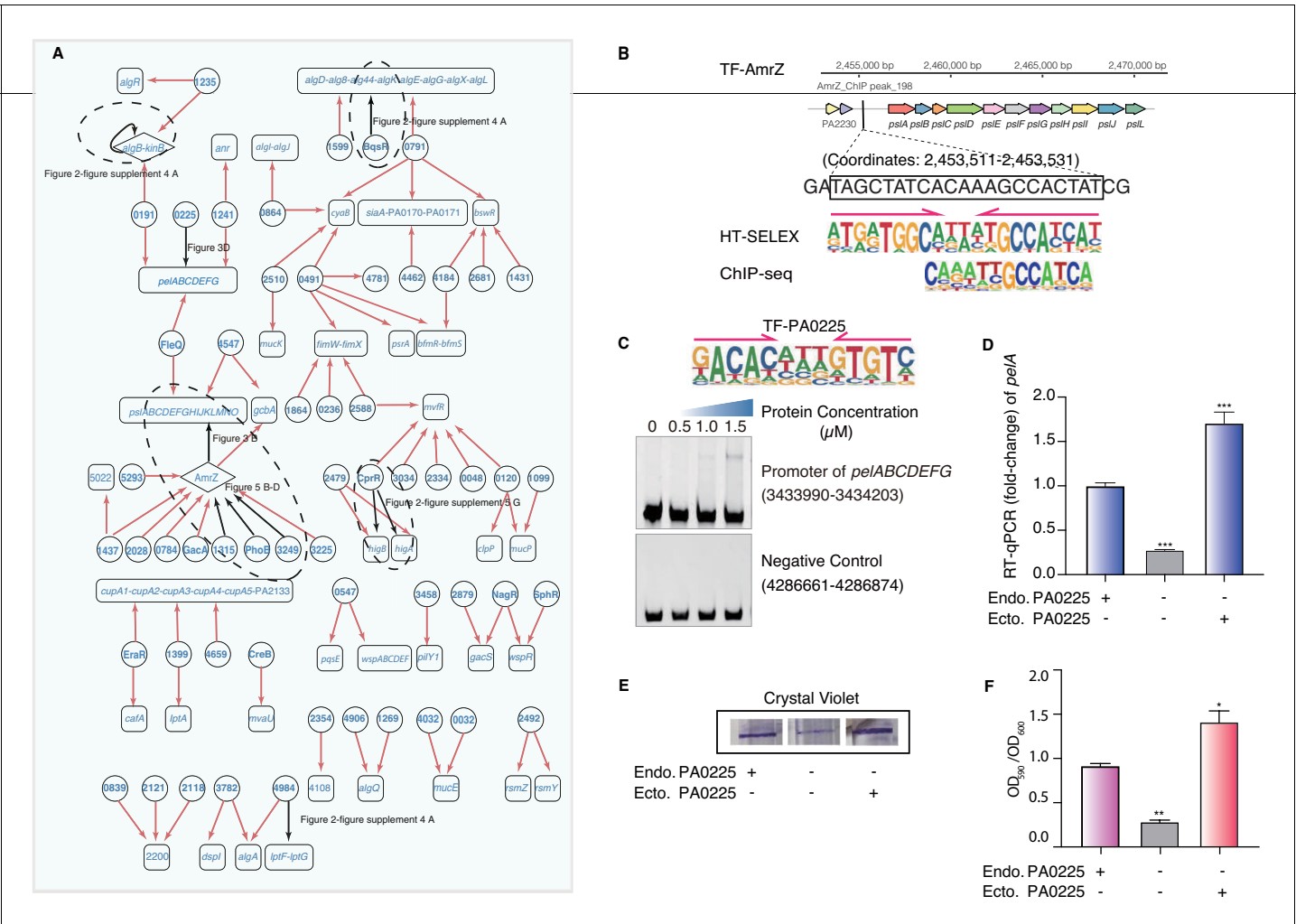

**Figure 3.** Transcriptional regulatory network in biofilm production pathway. (**A**) Network illustrates part of the regulatory relationship between TFs and their target genes in the biofilm pathway. Circles indicate TF proteins, and squares indicate target genes. Diamond highlights the gene with auto-regulation. Red arrows show that the binding sites are located in the putative promoters of the target genes, establishing the regulatory relationship. Black arrows highlight the binding with additional experimental validation. To save space, 'PA' is removed in the names of all TFs and their target genes. The dashed ovals and letters highlight regulatory relation validated in the corresponding figures and panels. (**B**) Comparison of binding motif of AmrZ derived by HT-SELEX ('SELEX') in the current study with a motif developed from a previous ChIP-seq study (upper). Arrows indicate half-sites in dimeric binding. Binding of AmrZ to a predicted site in the promoter of the *psl* operon is supported by a peak identified by a previous ChIP-seq study (lower) (***Jones et al., 2014***). (**C**) Electrophoretic mobility shift assay (EMSA) experiment validates the binding of PA0225, with its motif shown at the upper panel, to the promoter of *pelABCDEFG*. Arrows indicate half-sites in dimeric binding. Note that a binding-caused up-shift of the DNA band in gel is observed. By contrast, a negative control is used showing no binding to PA0225. (**D**) RT-qPCR shows that the transcription of *pelA* is significantly lower in PA0225 mutant cells compared with the wild-type cells (***p<0.001, Student's t-test). 'Endo. PA0225' indicates endogenous expression of PA0225. 'Ecto. PA0225' indicates the ectopic expression of PA0225 delivered by the transformed plasmid pAK1900. Three technical replicates were performed. (**E**) The biofilm formation detection in the wild-type, PA0225 mutant, and PA0225 complemented strains using a crystal violate staining assay. 'Endo. PA0225' indicates endogenous expression of PA0225. 'Ecto. PA0225' indicates the ectopic expression of PA0225 by the transformed plasmid pAK1900. (**F**) The quantification of biofilm production in the wild-type, PA0225 mutant, and PA0225 complemented strains using a crystal violate staining assay. 'Endo. PA0225' indicates endogenous expression of PA0225. 'Ecto. PA0225' indicates the ectopically expression of PA0225 by the transformed plasmid pAK1900. Three technical replicates were conducted (*p<0.05; **p<0.01, Student's t-test).

The online version of this article includes the following source data for figure 3:

**Source data 1.** Source data for *Figure 3D*.
**Source data 2.** Source data for *Figure 3F*.

2001), quorum sensing (QS) (*Lee and Zhang, 2015*), Type VI (*Ho et al., 2014*) and Type III secretion systems (T6SS and T3SS, respectively) (*Hauser, 2009*; *Hovey and Frank, 1995*; *Brutinel et al., 2008*), motility (*Coleman et al., 2020*), antibiotic resistance (*Chen et al., 2010*), siderophores (*Little et al., 2018*), stringent response (SR) and persistence (*Moradali et al., 2017*; *Goodman et al., 2004*; *Brauner et al., 2016*), and oxidative stress resistance (*Lan et al., 2010*). From the literature, we comprehensively summarized genes implicated in these nine virulence-associated pathways (*Supplementary file 5*), and then applied bedtools (*Wu and Jin, 2005*) to annotate all TFBSs that occurred in the promoter regions of genes involved in these pathways. Our prediction of the binding sites of most TFs in the *P. aeruginosa* genome using HT-SELEX and EMSA enabled us to systematically depict the transcriptional regulatory networks and identified key players of these important pathways. We carried out enrichment analysis of TF-binding sites located in promoter of genes involved in various functional pathways by the hypergeometric test: a TF was annotated association with a virulence-associated pathway when its binding sites were significantly enriched within promoters of genes in that pathway (FDR < 0.05). Consequently, we managed to associate 51 unique TFs with the nine virulence-associated pathways, 32 TFs of which had not been functionally characterized before (*Figure 2E*; *Figure 2—figure supplement 12*).

## PA0225 (ErfA) is a novel regulator of biofilm production

Biofilm is a complex community of diverse or single types of bacterial colonies embedded in an extracellular polymeric matrix that contains water, exopolysaccharides, extracellular DNA, proteins, and type IV pili (*Tseng et al., 2018*; *Burrows, 2012*; *Laventie et al., 2019*). *P. aeruginosa* produces biofilms to protect itself from host defense and antimicrobial agents, which helps enhance its pathogenicity (*Mulcahy et al., 2014*). We were interested in evaluating transcriptional regulation during biofilm production, which can be a critical step in development of strategies to control bacterial pathogenesis. To date, approximately 20 TFs have been reported to influence biofilm production (*Kang et al., 2017*; *Badal et al., 2020*; *Baraquet and Harwood, 2013*; *Hickman and Harwood, 2008*; *Baraquet et al., 2012*; *Dieppois et al., 2012*; *Parkins et al., 2001*; *Petrova and Sauer, 2009*; *Vallet et al., 2004*; *Monds et al., 2001*; *Ernst et al., 1999*; *Macfarlane et al., 1999*; *Ramsey and Whiteley, 2004*; *McPhee et al., 2006*; *Liang et al., 2008*; *Nicastro et al., 2009*; *Mikkelsen et al., 2009*; *Mukherjee et al., 2017*; *Finelli et al., 2003*; *Damron et al., 2009*; *Kong et al., 2015*; *Deretic and Konyecsni, 1990*; *Balasubramanian et al., 2012*; *Hammond et al., 2015*; *Petrova and Sauer, 2010*; *Petrova et al., 2011*; *Guragain et al., 2016*; *Sarkisova et al., 2005*; *Yang et al., 2019*; *Yeung et al., 2011*; *Zhao et al., 2016*). In our analysis, 57 putative TFs were predicted to regulate genes involved in biofilm production, including nine previously known regulators such as FleQ (*Baraquet and Harwood, 2013*; *Hickman and Harwood, 2008*), AlgB (*Chand et al., 2012*), and AmrZ (*Jones et al., 2014*; *Figure 3A*, *Figure 2—figure supplement 4*). Note that we were not able to recover any motif for the 11 other TFs that were known to be implicated in biofilm production. We reason that post-translational modification or protein–protein interaction may be required for their DNA binding.

Biofilm encases bacteria in exopolysaccharide, which is composed of the key polysaccharides Pel and Psl, among others (*Friedman and Kolter, 2004*; *Ryder et al., 2007*). In mammals, TFs are known to collaboratively bind to the regulatory elements in dense clusters (*Yan et al., 2013*; *Chronis et al., 2017*), and thereby synergistically regulate the transcription of important genes (*Hnisz et al., 2013*). Here, we observed a similar phenomenon of co-binding among the *P. aeruginosa* TFs. For instance, multiple TFs, including AmrZ, FleQ, and two uncharacterized TFs, PA5293 and PA4547, were predicted to bind to the same *psl* operon (*Figure 3A*, *Supplementary file 4*). Consistently, AmrZ was previously reported to bind to the *psl* operon and repressed its transcription, consequently decreasing biofilm production (*Jones et al., 2013*). A very strong AmrZ ChIP-seq peak was identified for the *psl* promoter containing our PWM-predicted binding site (Coordinates: 2,453,511–2,453,531) (*Jones et al., 2014*; *Figure 3B*), supporting the precise prediction of the binding site by our PWM motif. In addition, FleQ was previously observed to bind to the *psl* operon and consequently decrease biofilm production (*Hickman and Harwood, 2008*; *Jones et al., 2013*; *Baraquet and Harwood, 2016*). Our network confirmed the binding and suggested the precise binding site of FleQ in the promoter of the *psl* operon (*Figure 3A*).

In addition to the *psl* operon, we found that FleQ and three uncharacterized factors, PA0225 (ErfA) (*Trouillon et al., 2020*), PA0191, and PA1241, putatively bound the promoter of

another biofilm-associated operon *pelABCDEFG* (*Figure 3A*). The binding of FleQ to the promoter of *pel* clusters had been verified by a footprint assay in a previous study (*Baraquet and Harwood, 2016*). We aimed to verify the regulatory roles of the uncharacterized TFs in biofilm production, given their predicted binding to the promoter of the *pelABCDEFG* operon. We first confirmed the direct binding between PA0225 and the *pel* operon using EMSA (*Figure 3C*). Next, we generated a PA0225-deficient clone of *P. aeruginosa*. The deletion of PA0225 significantly decreased the transcriptional activity of *pelA* down to only one-third of that in wild-type (WT) cells. The depleted *pelA* expression could be completely restored by ectopic overexpression of PA0225 in the mutant cells (*Figure 3D*). These findings support the specific regulatory role of PA0225 in *pelA* transcription. It is known that *P. aeruginosa* biofilm production is sensitive to the cellular abundance of PelA (*Friedman and Kolter, 2004*; *Ryder et al., 2007*), and therefore we attempted to investigate whether the deletion of PA0225 could consequently lead to deficiency in biofilm formation, given that PA0225 was a potential regulator of *pelA* expression. We examined and quantified biofilm production in PA0225-knockout cells using a crystal violet staining assay. We found that the intensity of the crystal violet staining ring formed by the mutant cells was significantly lighter than that formed by WT *P. aeruginosa* cells. Like the *pelA* transcription, the cell adherence phenotype in the knockout cells could also be effectively recovered by overexpressing PA0225 via transformed pAK1900 plasmid, supporting the role of PA0225 in regulating biofilm production (*Figure 3E,F*).

To comprehensively validate the list of TFs putatively associated with biofilm production, we obtained transposon-mediated mutant strains of 57 putative biofilm-associated TFs (*Jacobs et al., 2003*) except PA2121, PA2118, and PA2028, due to lack of availability. Using a crystal violet assay, we confirmed that the depletion of known regulators such as AmrZ, FleQ, or PA0191 significantly affected biofilm formation (*Figure 2—figure supplement 4B*). Similar to other prokaryotic TFs, both activating and repressing phenotypes could be observed upon manipulating TF binding in *P. aeruginosa* (*Yang et al., 2019*; *Poole et al., 1996*; *Kawalek et al., 2019*; *Ha et al., 2004*; *Wu and Jin, 2005*; *Jin et al., 2011*; *Daddaoua et al., 2017*; *Chugani et al., 2001*; *Yan et al., 2019*). However, we observed that most mutant strains did not show an overt biofilm-associated phenotype, even some of those previously known biofilm-regulatory TFs such as RsaL (*Rampioni et al., 2009*), CprR (*Badal et al., 2020*), AlgB (*Chand et al., 2012*; *Mukherjee et al., 2019*), CdpR (*Zhao et al., 2016*), BqsR (Synonym: CarR) (*Guragain et al., 2016*; *Sarkisova et al., 2005*), and PA3782 (*Finelli et al., 2003*). We reasoned that unlike the knockout deletion that fully removed the gene products, the transposon insertion mutants could influence the expression of more than one gene because of the polar mutation effects on the expression of downstream genes, leading to less predictable phenotypes.

## TF-target network analysis reveals novel QS regulators

QS is a bacterial cell–cell communication system that fine-tunes the expression of hundreds of genes to produce, release, and recognize signaling molecules to monitor cell numbers and synchronize group behaviors (*Lee and Zhang, 2015*; *Williams and Cámara, 2009*; *Høyland-Kroghsbo et al., 2017*), while mediating cross-talks with a variety of virulence pathways involved in the aforementioned biofilm production and other important pathways such as T6SS, T3SS, motility, and antibiotic resistance (*Lee and Zhang, 2015*; *Williams and Cámara, 2009*). By using the PWM models of TFs to scan the genome, we predicted more than 100 TF-target gene relationships that occurred in the QS system, which involved a total of 49 TFs. In the putative QS network, four TFs had been previously annotated with regulatory roles in QS: Anr (*Hammond et al., 2015*), PsrA (*Wells et al., 2017*), PhoB (*Blus-Kadosh et al., 2013*), and RsaL (*Kang et al., 2017*; *Figure 4A*, *Figure 2—figure supplement 5A*). We predicted that an uncharacterized TF, PA1241, bound to the promoter of a QS-associated gene (*Figure 4A*). HT-SELEX generated a head-to-tail homodimeric motif for PhoB that was highly similar to a previously reported motif (*Bielecki et al., 2015*; *Figure 1C*). Our analysis showed that PhoB putatively conferred an auto-regulatory activity, with a predicted binding site on its own promoter (*Figure 4A*, *Supplementary file 4*). When we inspected the in vivo PhoB binding in the intergenic region upstream of its own transcription starting site, a very strong ChIP-seq peak was identified with its summit close to that of our PWM-predicted binding site (coordinate: 6,299,632–6,299,648) (*Bielecki et al., 2015*; *Figure 4B*).

Multi-TFs-mediated gene co-regulation was also commonly detected in the QS pathway. Co-binding sites of TFs were predicted in at least 10 promoters of QS-associated genes, including the

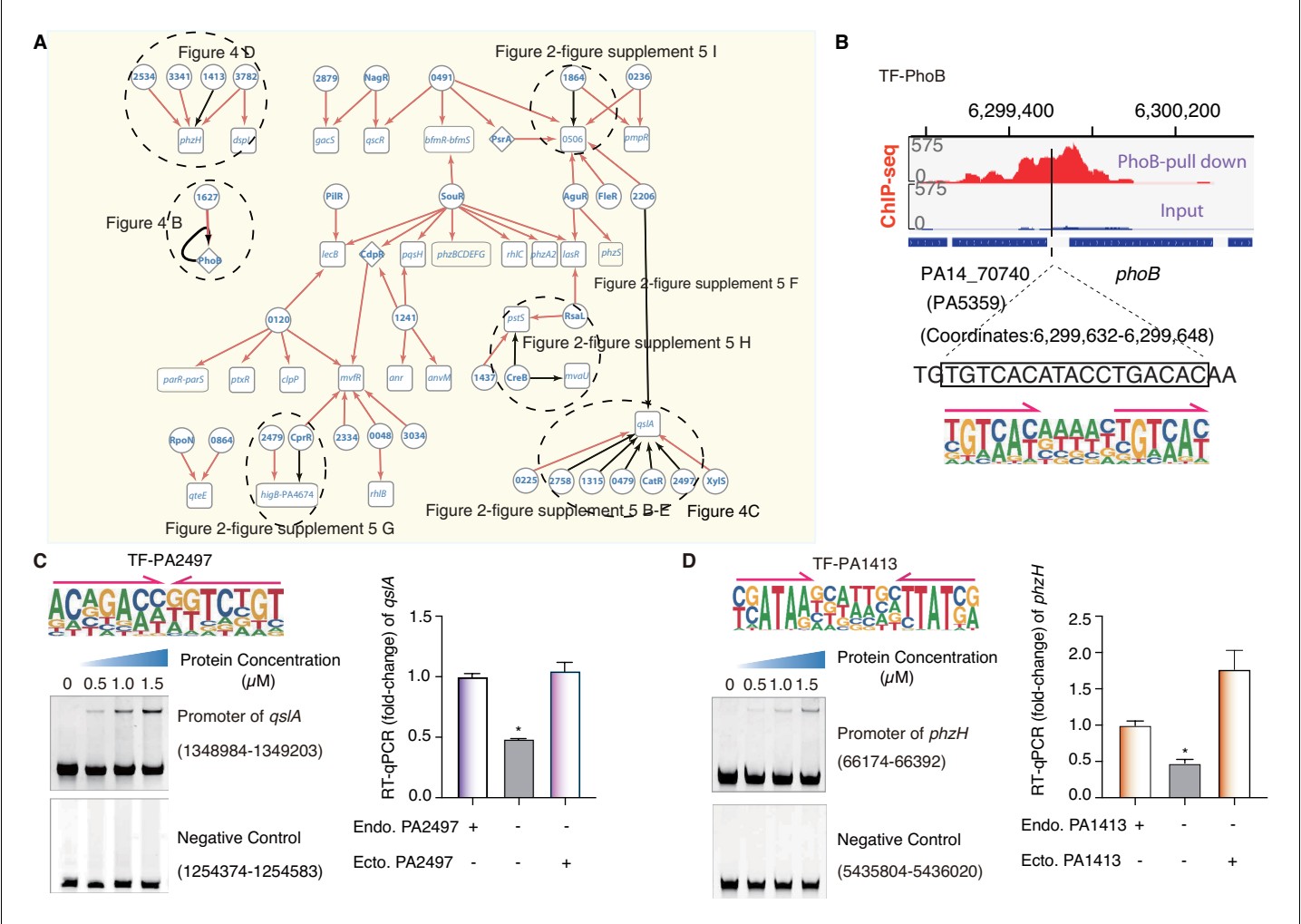

**Figure 4.** TF-target networks in QS pathway. (**A**) Network illustrates part of the regulatory relationship between TFs and their target genes in the QS pathway. The rest of the part is shown in *Figure 2—figure supplement 5*. Circles indicate TF proteins, and squares indicate target genes. Diamond highlights the gene with auto-regulation. Red arrows show that the binding sites are located in the putative promoters of the target genes, establishing the regulatory relationship. Black arrows highlight the binding with additional experimental validation. To save space, 'PA' is removed in the names of all TFs and their target genes. The dashed ovals and letters highlight regulatory relation validated in the corresponding figures and panels. (**B**) Arrows indicate half-sites in dimeric binding. Binding of PhoB to a predicted site in its own promoter, which was supported by a ChIP-seq peak. Input signal is shown for reference. The genomic coordinates are shown above the track and the genes (blue blocks) are indicated below the track. The exact binding site identified by our PhoB PWM is also highlighted. (**C**) Electrophoretic mobility shift assay (EMSA) and RT-qPCR validation of the predicted binding of PA2497 in the promoter of *qslA*. Left, EMSA experiment validates the binding of PA2497, with its motif shown at the upper panel, to the promoter of *qslA*. Arrows indicate half-sites in dimeric binding. Note that a binding-caused up-shift of the DNA band in gel was observed. By contrast, a negative control is used showing no binding to PA2497. Right, RT-qPCR shows that the *qslA* transcription is significantly lower in PA2497 mutant cells compared with the wild-type cells (*p<0.05, Student's t-test). 'Endo. PA2497' indicates endogenous expression of PA2497. 'Ecto. PA2497' indicates the ectopic expression of PA2497 by the transformed plasmid. Three technical replicates were conducted. (**D**) EMSA and RT-qPCR validation of the predicted binding of PA1413 in the promoter of *phzH*. Left, EMSA experiment validates the binding of PA1413, with its motif shown at the upper panel, to the promoter of *phzH*. Arrows indicate half-sites in dimeric binding. Note that a binding-caused up-shift of the DNA band in gel was observed. By contrast, a negative control is used showing no binding to PA1413. Right, RT-qPCR shows that the *phzH* transcription is significantly lower in PA1413 mutant cells compared with the wild-type cells (*p<0.05, Student's t-test). 'Endo. PA1413' indicates endogenous expression of PA1413. 'Ecto. PA1413' indicates the ectopic expression of PA1413 by the transformed plasmid. Three technical replicates were conducted.

The online version of this article includes the following source data for figure 4:

**Source data 1.** Source data for *Figure 4C*.
**Source data 2.** Source data for *Figure 4D*.

well-studied genes, for example *qlsA*, *phzH*, *lasR*, and *phoB*, and some previously uncharacterized genes, such as PA0506 and PA4674. In the QS pathway, LasR is a key regulator of the expression of more than 300 genes, while QslA is an anti-activator that interacts with LasR and prevents it from regulating its downstream targets (*Gilbert et al., 2009*; *Wade et al., 2005*; *Ueda and Wood, 2009*). The promoter region of the *qslA* gene was putatively co-occupied by eight different TFs (*Figure 4A*). EMSA results confirmed the binding of at least six TFs to this putative regulatory element, whereas the binding-caused gel shift was not observed for negative controls randomly selected from genomic loci without predicted binding sites (*Figure 4C*, *Figure 2—figure supplement 5B–F*). To further explore the regulatory role of uncharacterized TF binding to this promoter, we generated a clean deletion mutant of TF PA2497 that exhibited a significant decrease in the transcription of *qslA* compared to the WT strain. The deficiency could be effectively restored by ectopic overexpression of PA2497 in the mutant strain (*Figure 4C*). Similarly, TF PA1413 was predicted to bind to the promoter region of *phzH*, which controls the synthesis of the well-known QS-mediated virulence factor phenazine (*Mavrodi et al., 2001*; *Liang et al., 2011*). Our EMSA result verified PA1413 binding to the promoter region of *phzH* in vitro (*Figure 4D*). PA1413 deletion significantly reduced the transcriptional level of *phzH*, which could also be reinstated by ectopically expressing PA1413 in the knockout cells (*Figure 4D*).

## TFs involved in T6SS regulation

T6SS and T3SS are the two major types of effector protein-secreting apparatuses that contribute to strengthening the virulence of *P. aeruginosa*. T6SS is a bacteriophage-like membrane protein complex that injects T6SS effector proteins into target eukaryotic cells to cause host damage or delivers toxins into other prokaryotic cells to take control over inter-bacterial community competition, and is present in more than 200 Gram-negative bacterial species, including *P. aeruginosa* (*Hood et al., 2010*; *Russell et al., 2011*; *Russell et al., 2013*). We wired the interactions between TFs and T6SS-related genes and identified 37 TFs involved in the regulation of T6SS, 24 of which were previously uncharacterized (*Figure 5A*). Even for some of the previously known TFs, such as AmrZ, SphR, and SouR, we predicted new targets or new transcriptional regulatory relationships in the T6SS pathway. For example, TF SphR is responsive to host-derived sphingosine, which strengthens *P. aeruginosa* survival in mouse lungs (*LaBauve and Wargo, 2014*). Our data suggests that both SphR and the uncharacterized TF PA0048 bind to the promoter of the *tagJ1-tssE1-tssF1-tssG1-clpV1-vgrG1* operon, implying a cooperative function between SphR and PA0048 (*Figure 5A*).

There are three major clusters, H1-T6SS, H2-T6SS, and H3-T6SS, in the T6SS system (*Mougous et al., 2006*). AmrZ may play a key role in biofilm production, as we discussed earlier, through binding to the *psl* operon. AmrZ is also implicated in the T6SS pathway by suppressing the expression of H2-T6SS (*tssB2*) but activating that of H1-T6SS (*tssA1*) and H3-T6SS (*vgrG3*) (*Allsopp et al., 2017*; *Brencic and Lory, 2009*). Here, we observed that AmrZ might also control the transcription of the *vgrG1b* cluster genes *vgrG1b*, *tse6*, and PA0099 (*Figure 5A*), which is genetically associated with the H1-T6SS major cluster (*Pissaridou et al., 2018*). In addition to identification of their downstream targets, TFBSs of up to 10 TFs (GacA, PhoB, PA5293, PA0032, PA2028, PA1437, PA0784, PA1315, PA3249, and PA3225) were putatively found in the *amrZ* promoter region, which suggests a transcriptional regulatory cascade and potential inter-connection between different virulence pathways (*Figure 5A*).

Next, we verified the binding sites of TFs using multiple sets of data: the binding of PhoB to the *amrZ* promoter was supported by ChIP-seq data and confirmed in vitro by EMSA (*Figure 5B*); the binding sites of the TFs PA1315 and PA3249 were confirmed by EMSA using the genomic fragments covering the *amrZ* promoter (coordinate: 3,790,981–3,791,197) (*Figure 5C,D*); the binding site of TF PA0535 to the promoter of the H1-T6SS apparatus component-encoding gene *vgrG4* was also validated by EMSA (*Figure 5E*).

## Motility is under the control of 37 TFs

*P. aeruginosa* converts its biofilm lifestyle to a planktonic lifestyle by regulating bacterial motility, which enhances its infectivity of the host (*Drake and Montie, 1988*). Bacterial motility shares a few common regulators with biofilm production, including FleQ (*Arora et al., 1997*; *Jiménez-Fernández et al., 2016*), AlgB (*Chand et al., 2012*), and AmrZ (*Baynham et al., 2006*), but it has

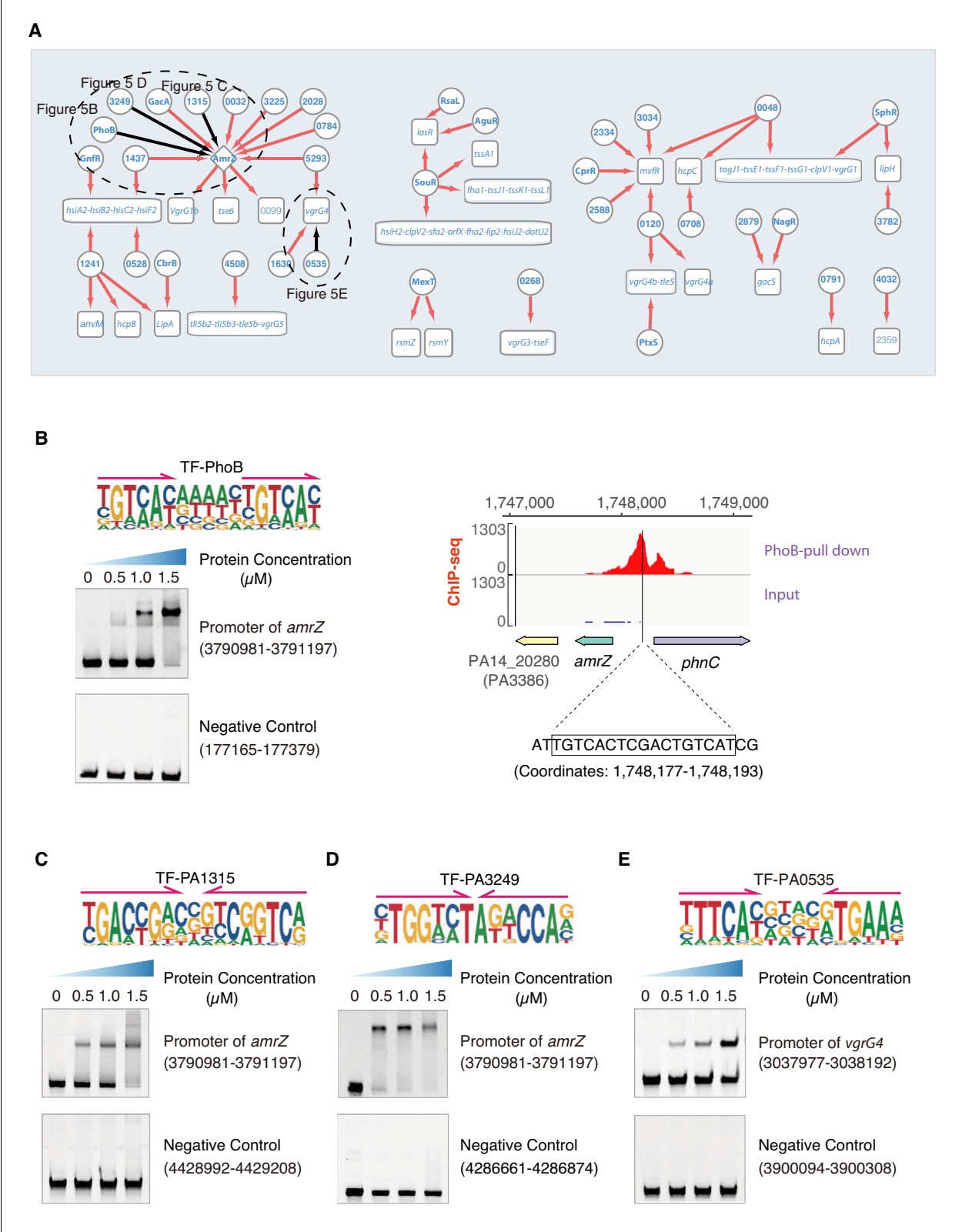

**Figure 5.** Transcriptional regulation in T6SS pathway. (**A**) Network illustrates the regulatory relationship between TFs and their target genes in T6SS pathway. Circles indicate TF proteins, and squares indicate target genes. Diamond highlights the gene with auto-regulation. Red arrows show that the binding sites are located in the putative promoters of the target genes, establishing the regulatory relationship. Black arrows highlight the binding with additional experimental validation. To save space, 'PA' is removed in the names of all TFs and their target genes. The dashed ovals and letters

*Figure 5 continued on next page*

*Figure 5 continued*

highlight regulatory relation validated in the corresponding figures and panels. (B) Left panel shows an electrophoretic mobility shift assay (EMSA) experiment to validate the binding of PhoB, with its motif shown at the upper panel, to the promoter of *amrZ* (left). Arrows indicate half-sites in dimeric binding. Note that a binding-caused up-shift of the DNA band in gel was observed. By contrast, a negative control was used showing no binding to PhoB. Right panel shows the binding of PhoB to a predicted site in the promoter of *amrZ*, which was supported by a ChIP-seq peak. Input signal is shown for reference. The genomic coordinates are shown above the track and the genes are indicated below the track. The exact binding site identified by our PhoB PWM is also highlighted. (C) EMSA experiment validates the binding of PA1315, with its motif shown at the upper panel, to the promoter of *amrZ*. Arrows indicate half-sites in dimeric binding. Note that a binding-caused up-shift of the DNA band in gel was observed. By contrast, a negative control was used showing no binding to PA1315. (D) EMSA experiment validated the binding of PA3249, with its motif shown at the upper panel, to the promoter of *amrZ*. Arrows indicate half-sites in dimeric binding. Note that a binding-caused up-shift of the DNA band in gel was observed. By contrast, a negative control was used showing no binding to PA3249. (E) EMSA validation of the predicted binding of PA0535, with its motif shown at the upper panel, to the promoter of *vgrG4*. Arrows indicate half-sites in dimeric binding. Note that a binding-caused up-shift of the DNA band in gel was observed. By contrast, a negative control was used showing no binding to PA0535.

many distinct TFs, such as GacA (*Brencic et al., 2009*) and PilR (*Kilmury and Burrows, 2018*). As expected, our network analysis reconfirmed the role of these TFs in regulation of motility-related genes (*Figure 2—figure supplement 6A*). For instance, FleQ was predicted to bind to the promoter of a flagellar gene *flhA*, and this prediction was supported by an independent study (*Jyot et al., 2002*). Similar to other pathways, we predicted many additional novel TFs involved in modulating bacterial motility. For example, PA3458 was predicted to bind to the promoter of the *fimU-pilV-pilW-pilX-pilY1-pilY2-pilE* operon, and PA1145, PA1399, PA3594, and PA4659 could all putatively bind to motility-related genes. Of these, PA3594 was predicted to bind the promoter of *flgBCDE*, which was confirmed by EMSA (*Figure 2—figure supplement 6B*). In sum, 37 TFs were predicted to regulate motility-related genes, in which 10 TFs were predicted to be highly involved regulators including RopN, PA0528, PA1145, PA1399, PA1490, EraR, PA3458, PA3594, PA4508, and PA4659 (*Figure 2—figure supplement 12*).

## TFs potentially involved in other virulence-associated pathways

Antibiotic tolerance in *P. aeruginosa* causes antibiotic treatment failure or infection relapse (*Brauner et al., 2016*). TFs influence antibiotic resistance by regulating multiple genes, including *mexF*, *mexE*, and *oprN*. The TF PhoB was predicted to influence antibiotic resistance by binding to the promoters of *pstB*, *oprD*, PA3516, and *czc*ABC, which was confirmed by both ChIP-seq (*Bielecki et al., 2015*) and EMSA (*Figure 2—figure supplement 7B–E*). Siderophores act as signaling molecules for the synthesis of two virulence proteins: exotoxin A and endo-proteinase PrpL (*Lamont et al., 2002*). The siderophore network revealed that 16 novel TFs could putatively bind to at least three siderophore-associated genes, with some loci co-bound by more than one TFs (*Figure 2—figure supplement 9*). Furthermore, 26, 17, and 11 TFs were predicted to be associated with the regulation of T3SS, ROS, and the SR and persistence pathways, respectively. Among them, 17, 11, and 8 factors, respectively, had not been functionally characterized before (*Figure 2—figure supplement 10*, *Supplementary file 4*).

Likewise, some TFs were putatively associated with multiple virulence-associated pathways. For example, PhoB and its target TF gene a*mrZ* were likely implicated in virtually all virulence-associated pathways, including biofilm production, QS pathway, T6SS, T3SS, antibiotic resistance, and motility. Similar findings were observed for other factors, such as PA0048, whose putative target genes were implicated in biofilm production (*mvfR*), the QS pathway (*rhlB*), T6SS (*tss*), antibiotic resistance (*pstB*), and the persistence pathway (*dnaJ*). Our results provide a valuable resource on an unprecedented scale to dissect the intricate transcriptional regulatory networks in different important biological processes in the pathogenic bacterium *P. aeruginosa* and explicitly illustrate the interconnectivity among these pathways.

## Gene ontology analysis reveals potential functions for 69 putative TFs

TFs exert their functions by binding to DNA and driving transcription of the target genes. To potentially decode the transcriptional regulatory function of all putative TFs, we performed a gene ontology (GO) enrichment analysis for the targets of each TF (*Figure 6*). The target genes of 38% (69/182) of the TFs were enriched for at least one GO term ($p<0.05$), and 21 functional categories were

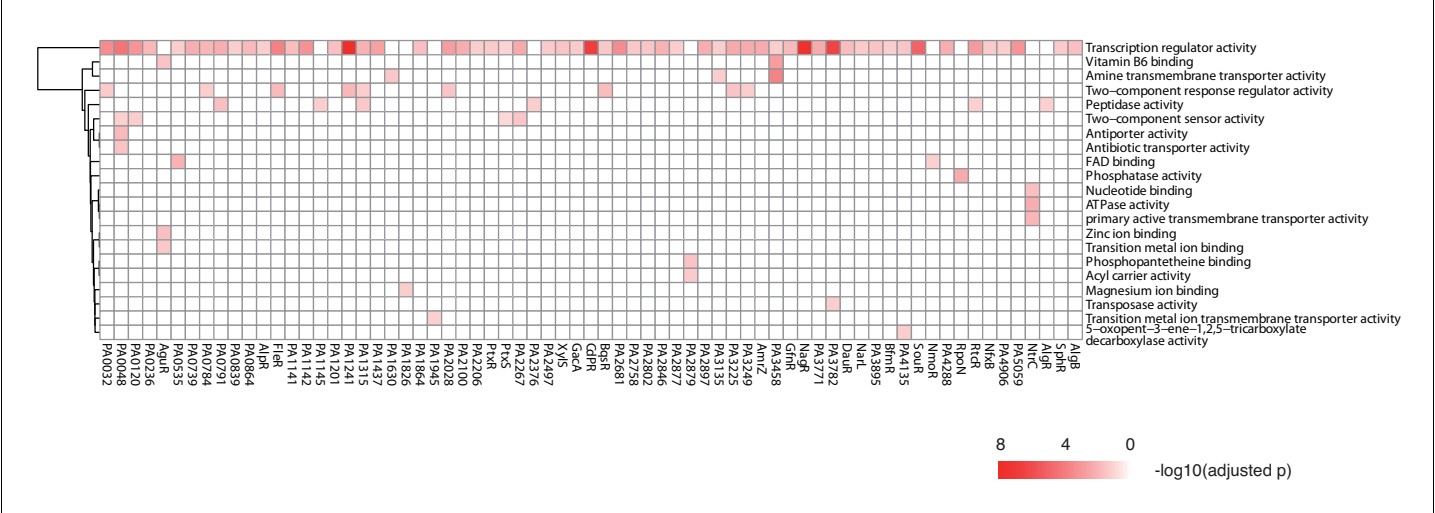

**Figure 6.** TF clustering by function of the putative targets. The heat map shows the functional annotation enrichment of TF targets in 29 gene ontology (GO) terms. The color key indicates the -log10 (p-value).

associated with the regulons of these TFs. As expected, virtually all of them were annotated with 'transcription regulator activity'. We then analyzed and explored the target genes for 20 TFs randomly selected from the 69 genes enriched for 'transcription regulator activity' and found that more than 80% of them coded for TFs (234/291), implying an inter-connective transcriptional regulatory network integrated with mutual regulation between TFs. In addition, the potential biological functions of 46 previously uncharacterized TFs were also suggested. For example, PA0048 could possibly be associated with 'antibiotic transporter activity', while the binding sites of PA3458 were enriched in promoters of genes with the 'vitamin B6 binding' and 'amine transmembrane transporter activity' characteristics. Interestingly, potentially additional functions were suggested for five previously annotated TFs, such as peptidase activity (AlgR and RtcR), FAD binding (NmoR), zinc ion binding (AguR), and phosphatase activity (RpoN).

## Discussion

Microbes appeared on earth nearly 2 billion years before the evolutionary arrival of human beings. Today, the population of microbes is estimated to be approximately 1 trillion, far exceeding the population of every other species in number and diversity (*Krause, 2002*; *Locey and Lennon, 2016*). High abundances of pathogenic microbes are found in high-density areas of human populations, especially in cosmopolitan urban centers (*Wang et al., 2018*). Due to the alarming emergence of drug-resistant pathogens across the planet, humans are being confronted with the unprecedented threat of untreatable infectious diseases (*Sattar, 2007*). However, very limited information and few useful treatment approaches are available to manage the infection and antibiotic resistance of such pathogens. *P. aeruginosa* is a model strain for studying pathogens as its virulence is controlled by many pathways. In this study, we successfully generated 199 unique PWM models to describe the DNA-binding specificities of 182 distinct TFs in *P. aeruginosa*, which covered nearly 50% of all TFs (including putative TFs) in its genome. We used all of the PWM models to scan the *P. aeruginosa* PAO1 genome and deduced 33,709 direct TF–DNA interactions, which enabled the construction of transcriptional regulatory networks for nine virulence-associated pathways, providing a valuable resource for deciphering the pathogenicity of *P. aeruginosa*.

Although many motifs obtained by HT-SELEX showed a high similarity to motifs derived from ChIP-seq peaks or other methods, such as sequence alignment, incomplete overlapping was observed for a few TFs (such as LasR) owing to method-specific differences, e.g. interference by protein–protein interactions, the number of TFBSs used to derived the motifs (*Figure 2—figure supplement 11A*). To further compare the PWM models between HT-SELEX and ChIP-seq, we used the PWM models of 10 TFs, namely AlgR, CdpR, ExsA, FleQ, GacA, MexT, PchR, PhoB, RsaL, and SphR,

to predict the binding events for each TF in the *P. aeruginosa* genome (*Huang et al., 2019*) and evaluated the performance with a precision-recall curve analysis. The result demonstrated that most TFs showed a satisfactory area under the precision-recall curve value of more than 0.5, demonstrating that our PWM could moderately predict the in vivo binding events (*Figure 2—figure supplement 11B*). ChIP-seq data are highly influenced by antibody quality and specificity, and the fact that a ChIP assay cannot distinguish between direct binding and indirect binding occurring between the protein and DNA. Owing to the small genome size, the binding motifs of TFs by ChIP-seq were likely skewed by the very limited number of motif-generating sequences. Consistent with the findings in higher species that TF-binding specificity is conserved during evolution, we also found that orthologous TFs from different strains displayed a similar binding specificity (*Fan et al., 2020*). For example, the DNA sequence preference of *P. aeruginosa* PhoB generated by HT-SELEX in our study showed a high similarity to that of *Caulobacter crescentus* PhoB generated by ChIP-seq (*Lubin et al., 2016*). This suggests that our data are a potential reference resource for TF studies in other related organisms.

Prior to this work, we and our collaborators studied the regulatory mechanisms of a group of *P. aeruginosa* TFs, including AlgR, CdpR, RsaL, VqsR, AnvM, and VqsM (*Kang et al., 2017*; *Kong et al., 2015*; *Zhao et al., 2016*; *Liang et al., 2012*; *Liang et al., 2014*; *Zhang et al., 2019*). By integrating those and other published data sets, we have created a *P. aeruginosa* genome-wide regulatory network (PAGnet), which illustrates the regulatory relationships of 20 key virulence-related TFs with their target genes as profiled by ChIP-seq and RNA-seq (*Huang et al., 2019*). Given that most TFs have yet to be characterized, we presume that the comprehensive regulatory network of *P. aeruginosa* would be more complicated than is currently known. The present work significantly contributes to the PAGnet by systematically predicting direct interactions between several more TFs and their target genes. Our data enables to envision the underlying transcriptional regulatory relationships and the investigation of the potential function of many previously uncharacterized regulators. The building of virulence models, which is a major step toward decoding pathogenicity, may lead to the discovery of novel drug targets for combating the infection of *P. aeruginosa* and other pathogens in the future.

## Materials and methods

### Strains and growth condition

The bacterial strains used in this study are listed in *Supplementary file 1A*. *P. aeruginosa* PAO1 WT strain, their derivatives, and *E. coli* strains were grown at 37°C in Luria-Bertani (LB) agar plates statically or LB broth with shaking at 220 rpm.

### Plasmids and primers

The plasmids and primers in this study are listed in *Supplementary file 1A*. Antibiotics for *E. coli* and its derivatives were used at the following concentrations: for *E. coli* with pET28a, 50 μg/ml kanamycin; for *E. coli* with pEX18AP, final concentration of 60 μg/ml ampicillin LB; for *E. coli* with pEX18AP-Gm plasmid, using final concentration of 15 μg/ml gentamycin in LB. For *P. aeruginosa* PAO1 with pEX18AP-Gm plasmid in LB media, antibiotic with final concentration 60 μg/ml gentamycin. For *P. aeruginosa* PAO1 with pAK1900 plasmid in LB media, antibiotic with final concentration of 100 μg/ml carbenicillin in LB. Antibiotics for *P. aeruginosa* PAO1 mutants, 60 μg/ml gentamycin.

### Cloning and recombinant protein purification

Oligonucleotides and vectors used for cloning of His-tagged proteins in this study are listed in *Supplementary file 2B*. The cloning was carried out with a homologous recombination strategy, following the manufacturer's instruction (Vazyme ClonExpress II One Step Cloning Kit, Vazyme Biotech). Briefly, the 371 TFs were identified in 'Pseudomonas Genome Database' (*Winsor et al., 2016*). DNA of 371 TFs were amplified by polymerase chain reaction (PCR) from *P. aeruginosa* PAO1 reference genome to obtain the coding regions of full-length proteins. Each forward PCR primer carried a 20 bp sequence identical to the linearized plasmid sequence at the 5'- and 3'-end of the cutting site followed by the gene-specific sequence. The homologous match between these two 20 bp recombination fragments determined the direction of the target gene in the expression

vector. Then the *BamH*I-digested pET28a vector and individual TF PCR products (containing 20 bp overlapped sequences on 5'- and 3'-end, respectively) were mixed in the molar ratio of 1:2, and then incubated with recombinase for 30 min at 37°C. Each successfully constructed vector (371 reconstructed vectors in total) was then transformed into *E. coli* BL21 (DE3) strain and cultured in the LB agar plate, respectively. Then, a single colony of each strain was picked and cultured into 3 ml LB overnight, which was transferred into 300 ml LB containing 50 µg/ml kanamycin for protein extraction. After bacterial $OD_{600}$ was near 0.6. 0.5 mM IPTG (isopropyl β-D-1-thiogalactopyranoside) was added into the cell culture with 16°C for 16 h. Then cell pellet was collected by centrifuging 7000 rpm for 5 min, at 4°C. The pellet was suspended in 15 ml buffer A (500 mM NaCl, 25 mM Tris-HCl, pH 7.4, 5% glycerol, 1 mM dithiothreitol, 1 mM PMSF (phenyl-methanesulfonyl fluoride)) and lysed by sonication for 30 min (20% power, 10 s on, 15 s off), and protein supernatant was obtained by centrifuging 12,000 rpm, 30 min, at 4°C. After filtering protein supernatant with a 0.45 µm filter, each filtrate was injected into a Ni-NTA column (Bio-Rad) to start a fast protein liquid chromatography (FPLC) system, respectively. The Ni-NTA column was washed with 30 ml gradient from 60 to 500 mM imidazole in buffer A gradually. Each eluted fraction was pooled, collected, and verified by sodium dodecyl sulfate-polyacrylamide gel electrophoresis (SDS-PAGE) based on molecular weight of each target TF plus 6× His-tag. HT-SELEX pipeline was exercised for all 371 TF proteins.

## HT-SELEX

The HT-SELEX experimental pipeline was adapted from our previous study (*Jolma et al., 2013*). Briefly, DNA ligand libraries that contain 40 bp randomized sequences with illumina or BGI adaptor systems were synthesized and double stranded by PCR amplification using the primers in *Supplementary file 1D*, illustrated in *Figure 1—figure supplement 1A–D*. The uniformity of each base (A, T, C, G) was evaluated after the sequencing of libraries by using Illumina HiSeq Xten or BGI MGISEQ 2000 sequencer. Then, 100–200 ng TF proteins and 5 µl DNA ligands were combined with Promega binding buffer (10 mM Tris pH 7.5, 50 mM NaCl, 1 mM DTT, 1 mM $MgCl_2$, 4% glycerol, 0.5 mM EDTA, 5 µg/ml poly-dIdC [Sigma P4929]) to make 25 µl of total volume. Protein–DNA complexes were incubated with 150 µl of Promega binding buffer (without poly-dIdC) containing 10 µl Ni Sepharose 6 Fast Flow resin (GE Healthcare 17-5318-01) equilibrated in binding buffer for 60 min with a gentle shaking after 30 min of reaction at room temperature. Subsequently, the unbound ligands were separated from the bound beads through washing with a gentle shaking for 12 times with 200 µl of Promega binding buffer (without poly-dIdC). After the washes, the beads were suspended by using 200 µl double-distilled water. Finally, the DNA libraries were built by 18 cycles of PCR amplification using the 20 µL bound DNA as template and the primers in *Supplementary file 1D*. The obtained PCR products were used as selection ligands for the next cycle of HT-SELEX. After the fourth cycle, the purified PCR samples from each HT-SELEX cycle were pooled and sequenced using Illumina HiSeq Xten or BGI MGISEQ 2000 sequencer.

## SELEX data analysis

Raw sequencing data were binned according to barcodes for each sample. Sequences from the 40-nt random region were further analyzed, the low quality reads with bases annotated as 'N' being filtered out. PWM models were generated using initial seeds identified using Autoseed (*Nitta et al., 2015*) that were refined by expert analysis as described in *Jolma et al., 2013*. Exact seeds, cycles, and multinomial model used are indicated in *Supplementary file 2*. All motif seqlogos were generated using the R package ggseqlogo (*Wagih, 2017*).

## Network analysis of similarity between PWMs

We calculated the similarities of all pairs of 199 PWMs using SSTAT (*Pape et al., 2008*) (parameters: 50% GC-content, pseudocount regularization, type I threshold 0.01) as described in *Jolma et al., 2013*. We generated a network containing two types of nodes: one type representing TF-binding profiles and another type representing TF proteins. TF protein nodes were connected to their binding models, and the binding models were further connected to each other if their SSTAT similarity score (asymptotic covariance) was greater than 1.5e-5 as described in *Jolma et al., 2013*. Finally, the network was visualized using Cytoscape software v3.7.2 (*Smoot et al., 2011*).

## Analysis of the transcriptional regulatory networks

We first comprehensively summarized nine virulence-associated pathways' gene lists based on literatures, and then scanned binding sites in *P. aeruginosa* PAO1 genome with all PWMs using FIMO (p<1.0e-5) (*Grant et al., 2011*). The p-value cutoff of 1.0e-5 predicted the binding sites that were closest to the peaks identified by ChIP-seq of PhoB which we had high-quality ChIP-seq data for. When we set the p-value cutoff of 1.0e-5, we obtained 22 binding sites, which is highly consistent with the ChIP-seq result. Therefore, we determined to use p<1.0e-5 as a standard cutoff for all FIMO prediction throughout our manuscript. Then bedtools (v2.25.0) (*Quinlan and Hall, 2010*) was used to annotate all TFBSs, which then intersected with the promoter regions of genes involved in nine virulence-associated pathways. Uncharacterised TFs involved in regulating the pathway genes are recognized as the pathway-associated TFs. We therefore generated transcriptional regulatory networks for nine important systems of *P. aeruginosa* which contained two types of nodes, one type representing TF proteins and another type representing targets. TF protein nodes were connected to their targets if the TFBSs were located in the promoter region of the targets. All networks were visualized using Cytoscape software v3.7.2 (*Smoot et al., 2011*).

## Electrophoretic mobility shift assay

EMSA is conducted in vitro using recombinant proteins and synthesized DNA ligands. DNA probes were PCR-amplified using primers listed in *Supplementary file 1A*, ranging from 210 bp to 240 bp long. Each PCR template was acquired from *P. aeruginosa* PAO1 genome. The 30 ng probe was mixed with varying amounts of TFs in binding buffer (10 mM Tris-HCl, pH 7.4, 50 mM KCl, 5 mM MgCl$_2$, 10% glycerol) with the final volume of 20 μl. Meanwhile, DNA probes of the negative controls in each group of reaction were chosen randomly and ensured without the corresponding TFBSs. After 30 min incubation at room temperature, the reactions were loaded and run by 6% polyacrylamide gel electrophoresis at 100 V for 1 hr. Then, the gels were subjected to nucleic acid dye for 5 min, and visualized and photographed using the gel imaging system (Bio-Rad). The assay was repeated at least twice with similar results.

## Construction of TF-deficient *P. aeruginosa* strains

Gene deletions were constructed as previously described (*Hoang et al., 1998*). The principle of gene knockout mutants depends on a SacB-based strategy. The pEX18AP plasmid was digested by using HindIII and EcoRI. The upstream arm (~1000 bp) and downstream arm (~1500 bp) of a TF gene were amplified from *P. aeruginosa* PAO1 genome and digested with XbaI (for detailed primers information, see *Supplementary file 1A*). Then the *Xba*I digested upstream and downstream fragments were ligated with T4 DNA ligase (NEB). The ligated DNA products were inserted into the *EcoR*I and *Hind*III digested pEX18AP plasmid using ClonExpress MultiS One Step Cloning Kit (Vazyme, China) to yield the pEX18AP-Up-Down plasmid. Then, pEX18AP-Up-Down of each TF was digested by XbaI and ligated with a 0.9 kb XbaI-digested gentamicin resistance cassette, generating pEX18AP-Up-Down-Gm plasmid, which was transformed into *P. aeruginosa* PAO1 WT competent cells with electroporation and cultured on the agar plate. Colonies were selected for gentamicin resistance and then transferred to LB agar plates containing 5% sucrose, which typically happens a double-crossover event and thus gene replacement. Each TF mutant was further confirmed by PCR to detect the DNA and RT-qPCR to detect the mRNA level.

## Reverse-transcription quantitative polymerase chain reaction (RT-qPCR)

*P. aeruginosa* PAO1 WT cells and its derivatives were cultured until OD$_{600}$ to 0.6. To harvest the bacterial cells, the cultures were centrifuged at 6000 rpm for 3 min. RNA extraction and purification were performed using RNeasy minikit (Qiagen) following the manufacturer's instruction. RNA concentration was measured by Nanodrop 2000 spectrophotometer (ThermoFisher). The synthesis of cDNA was carried out using the FastKing RT Kit (Tiangen Biotech). RT-qPCR was performed by SuperReal Premix Plus Kit (SYBR Green, Tiangen Biotech) following the manufacturer's instruction. Each reaction was performed in triplicates in 20 μl reaction volume with 20 ng cDNA and 16S rRNA as an internal control. For each reaction, 100 nM primers (*Supplementary file 1B*) were used for RT-qPCR. The reactions were run at the program on a PCR machine (42℃ for 15 min, 95℃ for 3 min, and then kept at 4℃). The fold change represents relative expression level of mRNA relative to the

16S rRNA control gene, which can be estimated by the values of $2^{-(\Delta\Delta Ct)}$. All the reactions were conducted with two biological repeats.

## Biofilm formation assay

Biofilm production was detected as previously reported in minor modifications (*King et al., 1954*). In brief, overnight bacterial cultures of *P. aeruginosa* PAO1 WT and TF mutants were transferred to a 10 ml borosilicate tube containing 1 ml LB medium (without antibiotics) with the original concentration $OD_{600} = 0.1$. Then, the cultures grow statically at 37°C for 12–24 hr. Then, 0.1% crystal violet was used to stain the biofilm adhered to the tube tightly for 30 min and other components bound to tube loosely was washed off with distilled deionized water (ddH$_2$O). Borosilicate tubes were washed for more than three times with ddH$_2$O gently, and the remaining crystal violet was fully dissolved in 1 ml 95% ethanol with constantly shaking after photograph. 100 µl of this eluate was transferred to a transparent 96-well plate to measure its optical density at 590 nm using Biotek microplate reader. The experiment was repeated using three independent bacterial cultures.

## GO analysis

GO enrichment analysis of TFs target genes was catalogued using DAVID version 6.7 (https://david-d.ncifcrf.gov/). The GO term with p-value < 0.05 was defined as significantly enriched term.

## Statistical analysis

Two-tailed Student's t-tests were performed using Microsoft Office Excel 2010. *p < 0.05, **p < 0.01, and ***p < 0.001 and error bars represent means ± standard deviation (SD). All experiments were repeated for at least twice. Statistical graphs were drawn using the GraphPad Prism 8, R ggplot2, and Python Matplotlib packages. All motif logos are drawn using R package ggseqlogo. All network was visualized by using Cytoscape software v3.7.2 (*128*).

# Acknowledgements

Funding: This work was supported by the National Natural Science Foundation of China (32070596 and 81873642 to JY, 31900443 to WS, 31870116 to XD), the Research Grants Council of Hong Kong SAR (21103018 and 11101619 to XD, 21100420 to JY), the City University of Hong Kong (9667188, 9610424 and 7005314 to JY), and the China Postdoctoral Science Foundation (2019M663794 to LF and 2019M663799 to WS).

# Additional information

## Funding

| Funder | Grant reference number | Author |
|---|---|---|
| National Natural Science Foundation of China | 8187364 | Jian Yan |
| National Natural Science Foundation of China | 31900443 | Wenju Sun |
| National Natural Science Foundation of China | 31870116 | Xin Deng |
| Research Grants Council, University Grants Committee | 21103018 | Xin Deng |
| Research Grants Council, University Grants Committee | 21100420 | Jian Yan |
| Research Grants Council, University Grants Committee | 11101619 | Xin Deng |
| China Postdoctoral Science Foundation | 2019M663799 | Wenju Sun |
| China Postdoctoral Science Foundation | 2019M663794 | Ligang Fan |

| City University of Hong Kong | 9667188 | Jian Yan |
| City University of Hong Kong | 7005314 | Jian Yan |
| National Natural Science Foundation of China | 32070596 | Jian Yan |
| City University of Hong Kong | 9610424 | Jian Yan |

The funders had no role in study design, data collection and interpretation, or the decision to submit the work for publication.

## Author contributions
Tingting Wang, Ligang Fan, Data curation, Validation, Investigation, Writing - original draft; Wenju Sun, Software, Formal analysis, Visualization, Writing - original draft; Canfeng Hua, Data curation, Investigation, Writing - original draft; Nan Wu, Shaorong Fan, Data curation, Validation; Jilin Zhang, Software, Writing - review and editing; Xin Deng, Conceptualization, Resources, Supervision, Funding acquisition, Investigation, Writing - original draft, Project administration; Jian Yan, Conceptualization, Supervision, Funding acquisition, Methodology, Writing - original draft, Project administration

## Author ORCIDs
Jian Yan https://orcid.org/0000-0002-1267-2870

## Decision letter and Author response
Decision letter https://doi.org/10.7554/eLife.61885.sa1
Author response https://doi.org/10.7554/eLife.61885.sa2

# Additional files
## Supplementary files
• Supplementary file 1. General information of the materials used in the study. (A) Strains, primers, and plasmids, related to *Figure 1*. (B) DBD family catalogues for 371 TFs, related to *Figure 1*, *Figure 1—figure supplement 1*.

• Supplementary file 2. Enriched motifs for HT-SELEX experiments, related to *Figure 1*.

• Supplementary file 3. The 104 distinct modules of the obtained PWMs for 182 TFs. Diamonds indicate TFs, and circles indicate individual PWMs. The dashed lines show the motif of the TF. The TFs without names are named with their locus tag omitting 'PA', related to *Figure 1B*.

• Supplementary file 4. Direct interactions between PWMs and their corresponding DNA sequences by scanning the genome-wide. TFBSs in the *P. aeruginosa* PAO1 reference genome using the FIMO software, related to *Figures 3–5*, *Figure 1—figure supplement 1*, and *Figure 2—figure supplements 1–12*.

• Supplementary file 5. List of TFs associated with nine known virulence pathways.

• Transparent reporting form

## Data availability
Sequencing data has been deposited in GEO under accession code GSE151518.

The following dataset was generated:

| Author(s) | Year | Dataset title | Dataset URL | Database and Identifier |
| --- | --- | --- | --- | --- |
| Yan J, Deng X, Wang T, Sun W, Fan L, Hua C | 2020 | An Atlas of the Binding Specificities of Transcription Factors in Pseudomonas aeruginosa Directs Prediction of Novel Regulators in Virulence | https://www.ncbi.nlm.nih.gov/geo/query/acc.cgi?acc=GSE151518 | NCBI Gene Expression Omnibus, GSE151518 |

The following previously published dataset was used:

| Author(s) | Year | Dataset title | Dataset URL | Database and Identifier |
|---|---|---|---|---|
| Bielecki P, Jensen V, Schulze W, Gödeke J, Strehmel J, Eckweiler D, Nicolai T, Bielecka A, Wille T, Gerlach RG, Häussler S | 2015 | Cross-regulation between the response regulators PhoB and TctD allows for the integration of diverse environmental signals in Pseudomonas aeruginosa | http://www.ncbi.nlm.nih.gov/geo/query/acc.cgi?acc=GSE64056 | NCBI Gene Expression Omnibus, GSE64056 |

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
