## [Decision Letter]

**Acceptance summary:**

This manuscript provides a global view of transcription factor interactions in *Pseudomonasaeruginosa* and represents an important resource for the research community. With the use of novel assays, the work has uncovered transcription factor binding specificities for nearly half of the annotated transcription factors and forms a good foundation for future studies. The study will be of interest to microbiologists and those interested in bacterial metabolism.

**Decision letter after peer review:**

Thank you for submitting your article "The Binding Specificity Atlas of *Pseudomonasaeruginosa* Transcription Factors Reveals Novel Regulators in Virulence" for consideration by *eLife*. Your article has been reviewed by 3 peer reviewers, and the evaluation has been overseen by a Reviewing Editor and Bavesh Kana as the Senior Editor. The following individual involved in review of your submission has agreed to reveal their identity: Yu Zhang (Reviewer #3).

The reviewers have discussed the reviews with one another and the Reviewing Editor has drafted this decision to help you prepare a revised submission.

Summary:

This manuscript analyzes the DNA binding sequence specificity of transcription factors (TFs) in the human pathogen *Pseudomonasaeruginosa* using a high-throughput DNA enrichment method. The consensus binding motifs of 182 (out of 371 in total) TFs were obtained and genomic loci were predicted to be targets of these TFs, some of which were experimentally validated. The work provides a useful resource for studying the transcription regulation network in *Pseudomonasaeruginosa*, and also provides DNA motif information for TFs in other bacteria. The reviewers all agree the work performed is generally of good quality, however, specific concerns raised by the reviewers regarding how the data are obtained, classified, and validated, need to be addressed.

Essential revisions:

1. One of the major conclusions is that most TFs bind as homodimers because their PWMs contain 'head-to-head' duplications of the same sequence. However, it could also be that two monomers bind a single double stranded DNA molecule in the HT-SELEX assays. It would be helpful if the authors could investigate whether homodimeric protein-protein interactions are known to occur for relevant TFs, or whether homodimers have been observed in structural studies.

2. The comparison of their HT-SELEX-derived PWMs with those obtained by ChIP should be expanded. After predicting binding events based on their PWM data, precision/recall analysis should be done with the ChIP data that compares individual binding events rather than the overall composite PWM binding motif. This will address the question of how many predicted binding events were also reported by ChIP and how many binding events detected by ChIP were also predicted. This is a key point.

3. The computational scanning for TF binding sites in the *Pseudomonas* genome is not sufficiently described: how was thresholding determined to call binding sites for each TF?

4. How a TF is associated with a biological pathway is not well described. How many binding events are required to call an association? Is this done one TF at a time, or one pathway at a time? How is a pathway defined?

5. The biological validation experiments appear ad hoc. It is not clear what this says about the entire dataset. For instance, only one new TF was validated experimentally out of 57 biofilm-associated TFs. The crystal violet assay can be easily performed in a 96-well plate format to validate 57 TFs that affect biofilm formation. PAO1 mutants can be obtained from Manoil lab (PAO1 transposon insertion library (https://www.gs.washington.edu/labs/manoil/libraryindex.htm)

6. It should be noted that the gel shift assays are similar to SELEX.

7. How many (predicted) binding events are functional? Analysis of (available) RNA-seq data would be key for this. For instance, Figure 1B. shows that the well-studied GacA and AmrZ have similar DNA binding specificity. Transcriptomic data are available for both these TFs (PMID: 31270321) and should be compared to the physical binding predictions.

8. Page 11. Lines 266-268, also Figure 5A. The authors state that ShpR shares six targets with PA4008 implying a cooperative (or redundant?) function in regulating T6SS. From the figure 5A, these 6 targets are: tagJ1, tssE1, tssF1, tssG1, clpV1, vgrG1. However these six genes are in the same operon and share a single promoter (https://*Pseudomonas*.com/feature/show/?id=102905&view=operons). The figure and the text imply that there are 6 independent TF binding sites located in 6 different promoters. However, there is a single promoter, and ShpR and PA4008 share only one target. Operon genes that share a single promoter are represented in the figures as independent genes. This is a major problem throughout the paper and figures and needs to be resolved.

9. Page 12. This paragraph is speculative as no functional motility assays were performed and no target genes were functionally validated. Thus the statement "in sum, 37 TFs were found involved (..) in regulating motility-related genes" is an overstatement.

10. Throughout the manuscript, the authors should be careful not to overstate their findings and be precise when biological function is predicted vs demonstrated experimentally.

11. The manuscript would benefit from thorough proofreading and editing.

12. Overall, 371 TFs entered in the pipeline and a result could be generated for 182 TFs (app. 50%). From an experimental design point of view, only 3 TFs have been used to benchmark the reproducibility of the binding site motifs which represents <1% of the input. This should be improved to provide a resource for the field. Since it is an in vitro assay, I would have expected to see the results validated systematically in duplicate. The author claim that the three replicates obtained "virtually identical binding specificity": but what does this mean statistically?

13. How did the authors select 182 TFs out of 371 TFs? Were there issues with protein purification for the others?

14. Line 167/Figure 2E: using the binding motifs, the authors predicted the involvement of 365 TFs in *Pseudomonas* virulence pathways. How is it possible to predict the role of 365 TFs while they obtained PWM for only 182 TFs?

15. Figure 3B/C: AmrZ binding site: it is not clear for the link between Figure 3B and 3C. Where is the motif in the psl operon?

16. The authors have already published a TF regulatory network paper (https://doi.org/10.1038/s41467-019-10778-w) covering 20 TFs : a supplementary figure that compares side-by-side the motifs found in the both papers could be interesting.

17. The author should explain why 201 sequence motifs (PWMs) were obtained for 182 TFs in the main text. For example, in the Data S1, the Module 37 of PA1241 has two sequence motifs containing almost the same consensus sequence. Why have two consensus sequence motifs have to be assigned?

18. The sequence motif for sig54 (rpoN) is conserved across bacteria and the core consensus sequence of GGN(10)GC has been confirmed in various studies. The sequence motif for sig54 by SELEX (Figure 1C) is inconsistent with previous literatures and should be reanalyzed and discussed.

19. The manuscript also describes interesting monomeric sequence motifs for 19 TFs. The discussion of these 19 TFs should be expanded.

20. The location of TF-binding sites relative to transcription start site (TSS) on the promoter DNA could help predict the activity of the corresponding TFs. For example, TF binding sites overlapping with the core promoter region (-35 to +1) and the proximal downstream region of a TSS may suggest transcriptional repression. As TSS profiles in *Pseudomonasaeruginosa* are published, it would strengthen the report to integrate the TSS information into the analysis herein.

[Editors' note: further revisions were suggested prior to acceptance, as described below.]

Thank you for resubmitting your work entitled "The Binding Specificity Atlas of *Pseudomonasaeruginosa* Transcription Factors Reveals Novel Regulators in Virulence" for further consideration by *eLife*. Your revised article has been evaluated by Bavesh Kana (Senior Editor) and a Reviewing Editor.

Summary:

This manuscript provides a global view of transcription factor interactions in *Pseudomonasaeruginosa* and represents an important resource for the research community. With the use of novel assays, the work has uncovered transcription factor binding specificities for nearly half of the annotated transcription factors and forms a good foundation for future studies. The study will be of interest to microbiologists and those interested in bacterial metabolism.

Although the manuscript has been improved, there are some remaining issues that need to be addressed, as outlined below:

1. There are still concerns about the interpretations of RpoN binding data. It is well established that RpoN doesn't function as a dimer to facilitate transcription initiation. It binds RNAP core enzyme in a 1:1 molar ratio to form an RNAP holoenzyme. Second, the sequence motif itself is not a direct repeat of two half-sites. The 5' 'GG' and 3' 'GC' consensuses of the GGN(10)GC motif are representative sequences of the '-24' and '-12' elements of a typical sigma 54 (encoded by rpoN)-initiated promoter. The two elements are recognized by two separate domains of RpoN, both in the presence and absence of RNAP holoenzyme. This issue of data interpretation and representation needs to be addressed. It is also recommended that the authors use a different example of head to tail binding.

2. The authors associate a TF with a pathway when a single predicted binding event occurs with any gene in that pathway. However, statistical analysis must be performed to determine the false discovery rate and whether their findings could occur via chance.

3. There are still points that the authors equate binding with function in the manuscript, please revise throughout to not make this overstatement.

---

## [Author Response]

Essential revisions:1. One of the major conclusions is that most TFs bind as homodimers because their PWMs contain 'head-to-head' duplications of the same sequence. However, it could also be that two monomers bind a single double stranded DNA molecule in the HT-SELEX assays. It would be helpful if the authors could investigate whether homodimeric protein-protein interactions are known to occur for relevant TFs, or whether homodimers have been observed in structural studies.

Thanks for the constructive comment. Many of the TF homodimers have been reported before by independent works, such as LasR (*1, 2*), RsaL (*3*), PsrA (*4, 5*), FleQ (*6*), and QscR (*7*). We have now cited these studies in our manuscript.

Furthermore, it should also be noted that unlike other cytosolic proteins, many TF dimer formations are mediated by sequence specific DNA molecules which they bound to and therefore may not be appropriately detected by protein-protein interaction assays. This has been shown before for both TF homodimers and heterodimers (*8, 9*). If the “head-to-head” motif had been indeed caused by “two monomers bind a single double stranded DNA molecule in the HT-SELEX assays”, the spacing between the monomer motif would have not been specific. In other words, two independent monomers could bind to a double stranded DNA in ANY orientation or spacing. Here in our study, we have shown that the TF homodimers displayed strong enrichment for particular orientation and spacing between two monomers. We have added the discussion to the revised manuscript. Please see Lines 154159.

2. The comparison of their HT-SELEX-derived PWMs with those obtained by ChIP should be expanded. After predicting binding events based on their PWM data, precision/recall analysis should be done with the ChIP data that compares individual binding events rather than the overall composite PWM binding motif. This will address the question of how many predicted binding events were also reported by ChIP and how many binding events detected by ChIP were also predicted. This is a key point.

Thanks for the suggestion. We agree that this is a very good way to evaluate the quality of our data. There are 10 TFs that were analyzed with both ChIP-seq (*10*) and HT-SELEX, including AlgR, CdpR, ExsA, FleQ, GacA, MexT, PchR, PhoB, RsaL, and SphR. We used their PWM models to predict the binding events for each TF in the *Pseudomonas aeruginosa* genome and assessed the performance with precision-recall analysis suggested by the reviewer. We found that most of TFs showed AUC (area under the precision-recall curve) over 0.5, demonstrating that our PWMs could well predict most of the in vivo binding (see Figure 2—figure supplement 11B). We also observed that prediction for some TFs is below 0.5, which is likely due to the fact that the binding is affected by chromatin accessibility, protein-protein interaction or other unknown mechanisms, which were quite common in higher species. The results are consistent with our direct comparison between motifs generated with ChIP-seq and HT-SELEX. We have discussed the comparison between ChIP-seq and PWM in the revised manuscript, please see Lines 419-429.

3. The computational scanning for TF binding sites in the Pseudomonas genome is not sufficiently described: how was thresholding determined to call binding sites for each TF?

Thanks for pointing out the missing information. The operation parameters of FIMO

(p<1.0E-5) to the manuscript, and other parameters take default values. This p value is the most commonly used cutoff when applying FIMO to scan TF binding sites in the genome. When we used the FIMO default p value cutoff (p<1.0E-4), an excessive number of peaks were predicted far beyond what we observed from the ChIP-seq data. For example, the highquality PhoB ChIP-seq detected 64 (34 of them were located upstream of a gene) binding peaks in vivo (*10*). When we set the p value cutoff of 1.0E-5, we obtained 22 binding sites, which is highly consistent with ChIP-seq result. Therefore, we determined to use p<1.0E-5 as a standard cutoff for FIMO prediction throughout our manuscript.

As the suggestion of the reviewer, we have expanded the Methods section by explicitly describing the threshold determination. Please see Lines 540-544.

4. How a TF is associated with a biological pathway is not well described. How many binding events are required to call an association? Is this done one TF at a time, or one pathway at a time? How is a pathway defined?

We apologize for the unclear description. We defined “association” as TF binding to at least one gene involved in a pathway or the TF itself was previously known to be implicated in the pathway. To predict direct interactions between TFs and DNA sequences, we first used the FIMO software (*11*) to scan genome-wide TF-binding sites (TFBSs) for the 182 TFs with available PWMs in the *P. aeruginosa* PAO1 reference genome (p<1.0E-5). Since the vast majority of binding within the gene body appeared to have little or no effect on transcription levels in prokaryotes according to previous studies (*12*), we primarily focused on predicted TFBSs in the intergenic regions in the subsequent analyses. In total, 33,709 significant TFBSs (p<1.0e-5) enriched in the intergenic regions were predicted. We are particularly interested in the transcriptional regulatory program in relation to the 9 virulence pathways of *P. aeruginosa*. Therefore, we comprehensively summarized 9 virulence-involved gene lists based on the last decades of studies, respectively. Then bedtools (v2.25.0) (87) was introduced to annotate all predicted TFBSs, which were intersected with the promoter regions of genes involved in these 9 virulence pathways. According to our analyses, TFs whose predicted binding sites were located in these genes are recognized as the corresponding pathway-associated TFs. We have revised the manuscript to include the explicit detail, please see Lines 189-195.

5. The biological validation experiments appear ad hoc. It is not clear what this says about the entire dataset. For instance, only one new TF was validated experimentally out of 57 biofilm-associated TFs. The crystal violet assay can be easily performed in a 96-well plate format to validate 57 TFs that affect biofilm formation. PAO1 mutants can be obtained from Manoil lab (PAO1 transposon insertion library (https://www.gs.washington.edu/labs/manoil/libraryindex.htm)

Thanks for suggesting the resource to evaluate our finding. We have obtained mutants and carried out the advised experiments. Among the 57 biofilm-associated TFs, 53 TF mutants were obtained from the Manoil lab except 4 mutants (PA0225, PA2121, PA2118, and PA2028). PA0225 has already been analyzed in our study while the mutant strains of PA2121, PA2118, and PA2028 were not available in their collection. A recent study had revealed that TF PA2121 functioned as a novel biofilm synthesis repressor in *P. aeruginosa* recently (*13*).

The crystal violet assay demonstrated that the biofilm production of the 10 TFs (PA0191, PA0491, PA0784, FleQ, PA1437, GacA, PA3249, AmrZ, SphR, and PhoB) had significant difference compared to the wild-type strain (P<0.05) (see Figure 2—figure supplement 4B), also consistent with the previous studies:

(1) FleQ, as c-di-GMP effector, positively regulates the expression of biofilm component- Pel exopolysaccharide in *P. aeruginosa* (*14-16*). Our crystal violet assay showed that FleQ mutant produced less biofilm compared to the wild-type, consistent to the previous study;

(2) AmrZ represses biofilm architecture by directly repressing transcription of the *psl* operon in *P. aeruginosa* (*17, 18*). Consistently, our crystal violet assay confirmed the point and the accurate binding site on the *psl* operon was successfully identified by AmrZ PWM generated from HT-SELEX (Coordinates: 2,453,511-2,453,531) (see also Figure 3B);

(3) GacA is a regulatory element in *P. aeruginosa* biofilm formation. A *gacA* mutant contains a deficit biofilm development revealed by scanning electron microscopy (*19*), consistent to our crystal violet result;

(4) PhoB negatively modulates the biofilm production of *P. aeruginosa* by controlling the Pho regulon (*20*), consistent to our crystal violet assay result that the *phoB* mutant strain produced more biofilm production compared to the wild-type strain.

Although the altered biofilm production of the remaining 43 TFs mutant strains were not significant in our assay, 6 TFs, including RsaL (*21*), CprR (*22*), AlgB (*23, 24*), CdpR (*25*), BqsR (Synonym: CarR) (*26, 27*), and PA3782 (*28*) had been reported to influence biofilm production in previous studies.

(1) The *rsaL* mutant is impaired in biofilm formation (*21*);

(2) CprR is involved to biofilm formation on indwelling medical devices such as endotracheal tubes (ETTs) (*22*);

(3) AlgB is required for the repression of colony biofilm formation by repressing biofilmrelated genes, while the *algB* mutant had a colony biofilm phenotype indistinguishable from the wild-type (*24*);

(4) A ∆*cdpR* mutant results in increased production of biofilm (*25*);

(5) BqsR (or named CarR) modulates Ca^2+^ signaling to influence biofilm production (*26, 27*); (6) The inactivation of PA3782 caused defects in biofilm formation in static and flowing systems (*28*).

We reasoned that unlike the clear deletion, the transposon insertion mutants could influence the expression of more than one genes because of the polar effects on the expression of downstream genes, or that mutating one TF gene could not change the biofilm phenotype owing to the complexity of regulation in vivo, e.g. the case of AlgB mentioned above.

6. It should be noted that the gel shift assays are similar to SELEX.

Thanks for the advice. We have noted that EMSA and HT-SELEX are similar in that they are both conducted in vitro using recombinant proteins and synthesized DNA ligands in the revised manuscript. However, we would like also to highlight the fundamental difference between these two experiments: 1) HT-SELEX introduces a large pool of DNA sequences for competition and selection, while EMSA can only analyzed one sequence at a time; 2) HTSELEX is based on exponential enrichment of binding sequences along the repeated cycles while EMSA is less quantitative in terms of binding affinity; 3) HT-SELEX can be easily adapted for high-throughput study, investigating binding of multiple TFs with multiple DNA sequences, while EMSA is difficult to be carried out in a high-throughput manner. Therefore, EMSA could be used as an orthogonal method to validate the individual binding events. In addition to EMSA, we also used other experiments, such as ChIP-seq to validate our findings.

7. How many (predicted) binding events are functional? Analysis of (available) RNA-seq data would be key for this. For instance, Figure 1B. shows that the well-studied GacA and AmrZ have similar DNA binding specificity. Transcriptomic data are available for both these TFs (PMID: 31270321) and should be compared to the physical binding predictions.

We appreciate the reviewer for suggesting multiple ways of validating the quality of our HTSELEX data. Accordingly, we have downloaded Huang et al. (*10*) RNA-seq data to compare the overlapping of the changed expression upon deletion. The list of GacA and AmrZ manipulation caused differentially expressed genes (defined as > 2 times fold change of the expression) was intersected with the list of GacA and AmrZ target genes predicted in this paper respectively (see Author response image 1). The results showed that only a small fraction of predicted target genes were dysregulated upon knockdown of the TF in cells. This could be due to several factors: 1) many TFs co-bound to the promoter region and therefore loss of one TF binding was not sufficient to significantly change the expression of the following gene by twofold; 2) Huang et al. did RNA-seq at the log phase (OD=0.6) only, whereas other growth phases could show a very different list of DEGs; 3) complex regulatory mechanism existed for the transcription, involving protein-protein interactions, chromatin accessibility, so that the changed transcription did not reflect cellular response to a simple one-to-one relationship between TF and target gene. Predicting transcription output is still a very challenging task in the field, even if we integrate the information of chromatin epigenetic information, protein abundance, and 3D genome folding, etc. Therefore, we avoided using the TF depletion followed by RNA-seq to assess the PWM quality in our manuscript. We have included the discussion in the revision.

**Author response image 1. respfig1:** Venn diagrams show the comparison of predicted GacA (left) or AmrZ (right) target genes (red) and differentially expressed genes upon deletion of GacA or AmrZ (blue).

8. Page 11. Lines 266-268, also Figure 5A. The authors state that ShpR shares six targets with PA4008 implying a cooperative (or redundant?) function in regulating T6SS. From the figure 5A, these 6 targets are: tagJ1, tssE1, tssF1, tssG1, clpV1, vgrG1. However these six genes are in the same operon and share a single promoter (https://Pseudomonas.com/feature/show/?id=102905&view=operons). The figure and the text imply that there are 6 independent TF binding sites located in 6 different promoters. However, there is a single promoter, and ShpR and PA4008 share only one target. Operon genes that share a single promoter are represented in the figures as independent genes. This is a major problem throughout the paper and figures and needs to be resolved.

Thanks a lot for pointing out this issue and the constructive suggestion. To avoid confusion, we have revised the sentence to emphasize the point in manuscript, please see Lines 327 to 333. We also have merged all the genes sharing the same operon into one box in the network illustration to avoid the confusion. Please see revised Figure 3, 4 and 5 and Figure 2—figure supplement 4, 5, 6, 7, 8, 9, 10.

9. Page 12. This paragraph is speculative as no functional motility assays were performed and no target genes were functionally validated. Thus the statement "in sum, 37 TFs were found involved (..) in regulating motility-related genes" is an overstatement.

Thanks a lot for pointing out this issue. To avoid overstatement and confusion, we revised the statement from “In sum, 37 TFs were found involved in regulating motility-related genes.” to “In sum, 37 TFs were predicted to regulate motility-related genes.” Please see Line 358-359.

We have also gone through the manuscript and tuned down the overstatement of our findings.

10. Throughout the manuscript, the authors should be careful not to overstate their findings and be precise when biological function is predicted vs demonstrated experimentally.

Thanks a lot for pointing out the overstatement issue in this manuscript. We have carefully checked the manuscript and clearly differentiated predicted and experimentally verified TFBS to avoid confusion.

11. The manuscript would benefit from thorough proofreading and editing.

Thanks a lot. A scientist who is a native speaker of English has carefully checked the grammatical issues and edited the text throughout the manuscript.

12. Overall, 371 TFs entered in the pipeline and a result could be generated for 182 TFs (app. 50%). From an experimental design point of view, only 3 TFs have been used to benchmark the reproducibility of the binding site motifs which represents <1% of the input. This should be improved to provide a resource for the field. Since it is an in vitro assay, I would have expected to see the results validated systematically in duplicate. The author claim that the three replicates obtained "virtually identical binding specificity": but what does this mean statistically?

According to our previous experience, HT-SELEX is a highly quantitative and reproducible method. In our previous publication (*8*), we have systematically compared the technical replicates (replicates using the same protein but different DNA input libraries) and biological replicates (replicates using different protein clones, including DNA-binding domains and full length proteins of the same TFs) and demonstrated that HT-SELEX resulted in virtually the same results. Therefore, we only picked up three TFs as QC controls when we carried out HT-SELEX in different batches and expectedly these replicates turned out identical motifs.

To validate the results, we also used orthologous methods, e.g. we carried out hundreds of EMSA to validate individual binding sites predicted by PWM and also showed the consistency between HT-SELEX generated motifs and ChIP-seq generated motifs for some TFs for which both data are available.

In order to demonstrate the reproducibility in an acceptable timeframe of revision, we randomly chose 13 TFs and carried out HT-SELEX with a different batch protein expression and newly synthesized input oligos, and also with a different experimental researcher. Of the 13 TFs, nine TFs showed explicit motifs in our first batch of experiment while the other 4 did not. In the second batch of experiment, we successfully obtained identical motifs for all the 9 TFs (see Author response image 2).

**Author response image 2. respfig2:** A table shows comparison of motifs discovered in different replicative experiments of TF HT-SELEX. Note that the proteins and input DNA were both independently synthesized and all motifs for the same TF were identical.

For the other 4 TFs that we were not able to observe sequence specificity (GbdR, SoxR,

VqsR and QscR), we did not recover any motif either. We then checked the protein expression and found that except QscR, the other TF proteins were very abundant, suggesting that failure of motif enriched was unlikely to be caused by lack of protein in the reaction (see Author response image 3). The same question was also discussed below in response to the next point of reviewer’s comment.

**Author response image 3. respfig3:** SDS-PAGE results detecting the TF protein expression. The three TFs, GbdR, SoxR, and VqsR showing abundant expression did not recover any specific DNA binding motifs.

13. How did the authors select 182 TFs out of 371 TFs? Were there issues with protein purification for the others?

We apologize for confusing the reviewer due to our poor wording. We actually did not “select” any TFs. We have performed HT-SELEX for all 371 TFs, out of which 182 TF showed DNA binding specificities. We don’t think the failure was caused by protein purification issue as we carefully inspected protein amount for each HT-SELEX experiment. We ran SDS-PAGE gel for all TF proteins used in our paper (371 TFs) and measured the concentrations carefully. We used 100-200 ng per TF in each round of SELEX. The enrichment of specific motifs is not related to the protein abundance. For example, PA3122, PA3260 and PA3932, have a clear and bright protein band in SDS-PAGE gel, but HTSELEX did not detect any specific DNA sequence motifs (see Author response image 4).

**Author response image 4. respfig4:** SDS-PAGE gel result for 23 purified TFs (Left). All TFs expressed successfully. Only 13 TFs had HT-SELEX-generated motifs.

14. Line 167/Figure 2E: using the binding motifs, the authors predicted the involvement of 365 TFs in Pseudomonas virulence pathways. How is it possible to predict the role of 365 TFs while they obtained PWM for only 182 TFs?

As stated above, we apologize for the poor wording that caused confusion. The 365 TFs included redundant counting for TFs involved in multiple different pathways. Of these 365 TFs, only 127 TFs are unique, among which 92 TFs had never been characterized before. We have revised the manuscript, and noted the number clearly. Please see Lines 33-34, and 199200 in the revised manuscript.

15. Figure 3B/C: AmrZ binding site: it is not clear for the link between Figure 3B and 3C. Where is the motif in the psl operon?

The binding site “TAGCTATCACAAAGCCACTAT” was identified by scanning the *Pseudomonas aeruginosa* PAO1 genome with the PWM model of AmrZ generated in this study (FIMO score 8.75E-06). To clarify, we have now merged the former Figure 3 C with Figure 3 Band highlighted the predicted AmrZ-binding site.

16. The authors have already published a TF regulatory network paper (https://doi.org/10.1038/s41467-019-10778-w) covering 20 TFs : a supplementary figure that compares side-by-side the motifs found in the both papers could be interesting.

Thank you. This is a similar question to Point 2 above. We have provided a supplementary table (Figure 2—figure supplement 11A) to compare the ChIP-seq motif and HT-SELEX motif side-by-side, in total 13 TFs with both datasets available, as suggested by this reviewer (see Author response image 5). By comparison, 11 TFs shared identical or similar motifs derived from the two methods, including PhoB, RsaL, AmrZ, LasR, FleQ, PchR, ExsA, AlgR, CdpR, GacA and SphR. For BfmR, ChIP-seq didn’t recover any motif while HT-SELEX reported two weak but similar motifs (TANNNTA or TANNNNNNNNTA). As we discussed above, these two methods are used to investigate different problems, conducted in different experimental settings: ChIP-seq is mainly used to survey the in vivo binding sites which could be affected by protein-protein interactions, genome accessibility, chromatin 3D organizations (enhancer-promoter loop), etc while HT-SELEX primarily focused on biochemical binding affinity between TF protein and DNA sequences. In our previous studies, we have shown that in many cases they were not completely identical (*29*). We have added the comparison data and a brief discussion to the revised manuscript. See Lines 419438.

17. The author should explain why 201 sequence motifs (PWMs) were obtained for 182 TFs in the main text. For example, in the Data S1, the Module 37 of PA1241 has two sequence motifs containing almost the same consensus sequence. Why have two consensus sequence motifs have to be assigned?

Thanks for the advice. Some TFs bind to the DNA in a homodimeric mode with different spacing and monomer orientation (*8*). Therefore, multiple PWMs were assigned to one TF to respectively describe its different spacing and orientation. This has been discussed before in our previous studies (*8, 30*). To avoid confusion, we have noted this in the revised manuscript. Please see Lines 91-96.

18. The sequence motif for sig54 (rpoN) is conserved across bacteria and the core consensus sequence of GGN(10)GC has been confirmed in various studies. The sequence motif for sig54 by SELEX (Figure 1C) is inconsistent with previous literatures and should be reanalyzed and discussed.

We deeply appreciate this reviewer for pointing out a potential issue of the RpoN PWM we reported in the first submission. In our first round of HT-SELEX, we have identified a motif that was different from conventional and highly conserved RpoN motif reported for different bacterial species, including *Vibrio cholerae*, *Salmonella enterica Serovar Typhimurium*, *Yersinia pseudotuberculosis, Agrobacterium tumefaciens, Geobacter sulfurreducens, etc.* Then, we repeated the HT-SELEX experiment for RpoN using independently purified protein lysates and input DNA oligos for 4 times, we have now detected the GGN(10)GC conventional motif as the most enriched one (see Author response image 5), although the unknown RpoN motif was still observed in one of the replicative experiment. By carefully inspecting this unknown motif, we found that it might result from a mixed mode of two tandem half sites (TGC) on the 3’-side while the 5’-side half site (GGC) was missing. We can also see that the 5’-half site sometimes shows much weaker specificity, e.g. in both replication 1 and replication 3, compared to 3’-half site. So we reason that the folding of RpoN proteins was sometimes flexible in binding to DNA, which gives more DNA binding flexibility when it forms homodimer or heterodimers with other proteins. We have added discussion on RpoN binding in revised manuscript, Line 124-130: “We reason that RpoN proteins could sometimes form dimers while binding to DNA, and in this form, the 5′-side half site binds to another protein molecule. This result demonstrates that HT-SELEX is a highly sensitive method for detecting any change in protein topology.”

**Author response image 5. respfig5:** Table shows comparison of RpoN motifs. Original motif shows the PWM logo of RpoN binding we obtained in original submission while the current motifs shows the replicative experiments in revision. Note that we repeated the HT-SELEX for 4 times with different input DNA ligands. Except in the replication 1 that we detected both conventional RopN motif and the motif we obtained in first batch, the other 3 replicates only detected the conventional RopN motif.

19. The manuscript also describes interesting monomeric sequence motifs for 19 TFs. The discussion of these 19 TFs should be expanded.

LysR and AraC are the top 2 families with the largest TFs in *Pseudomonas aeruginosa*

PAO1. Among 182 TFs with successful SELEX motifs, 43% belongs to LysR and AraC family (see Author response image 6). In general, the monomeric sites are shorter and more frequent in the genome, indicating their broad regulatory roles of these TFs. Meanwhile, we also observed that there were also prevalent homodimer binding modes for TFs belonging to these two families in addition to these monomeric TFs. The availability of both monomer and homodimer binding modes reflected the diverse amino acid sequences for DNA binding domains of TFs in these two families and could be used for further classification of these TFs. We have noted the broad target sites and the high diversity of DNA binding specificities for these TFs in the discussion. Please see Lines 147 to 151 in the revised manuscript.

**Author response image 6. respfig6:** 

20. The location of TF-binding sites relative to transcription start site (TSS) on the promoter DNA could help predict the activity of the corresponding TFs. For example, TF binding sites overlapping with the core promoter region (-35 to +1) and the proximal downstream region of a TSS may suggest transcriptional repression. As TSS profiles in *Pseudomonas aeruginosa* are published, it would strengthen the report to integrate the TSS information into the analysis herein.

Thanks for your advice. We also agree that it would be very useful information to annotate the genome with the location of TF-binding sites relative to transcription start site (TSS) on the promoter. Unfortunately, the published TSS information of *Pseudomonas aeruginosa* corresponded to the UCBPP-PA14 strain (*31*) which was different from the *Pseudomonas aeruginosa* PAO1 strain used in this study. We could not find a proper tool to liftover the two reference genomes and therefore to avoid confusing future users, we chose not to include such information. Researchers may have their own TFBS prediction in the PA14 strain (using PWM from the current study) to associate the TFs with corresponding target genes if interested.

To compare the location of TFs binding sites relative to TSS in the PAO1 strain, we illustrated several examples in Author response image 7. Based on the TSS information originated from PA14, we deduced the corresponding TSS of PAO1 by sequence alignment. We have now added the TSS information (*31*) of *pel* operon (see Author response image 7). For example, PA0225 and PA2497 were predicted to bind to the upstream region of *pelA* and *qslA* TSS, respectively. Our RTqPCR and crystal violet results validated that PA0225 positively regulated the expression of *pelA*. Similar cases, such as PA2497, PA1241, and PA1234 were predicted to bind to the upstream region of *qslA, anvM,* and *algB* TSS, respectively (see Author response image 7).

**Author response image 7. respfig7:** The association between predicted TSS and TF binding sites for PA0225 and PA2497.

References:

1. M. J. Bottomley, E. Muraglia, R. Bazzo, A. Carfi, Molecular insights into quorum sensing in the human pathogen *Pseudomonas* aeruginosa from the structure of the virulence regulator LasR bound to its autoinducer. *J Biol Chem* 282, 13592-13600 (2007).

2. H. Fan *et al.*, QsIA disrupts LasR dimerization in antiactivation of bacterial quorum sensing. *Proc Natl Acad Sci U S A* 110, 20765-20770 (2013).

3. H. Kang *et al.*, Crystal structure of *Pseudomonasaeruginosa* RsaL bound to promoter DNA reaffirms its role as a global regulator involved in quorum-sensing. *Nucleic Acids Res* 45, 699-710 (2017).

4. Y. Kang *et al.*, The long-chain fatty acid sensor, PsrA, modulates the expression of rpoS and the type III secretion exsCEBA operon in *Pseudomonas* aeruginosa. *Mol Microbiol* 73, 120-136 (2009).

5. M. Kojic, C. Aguilar, V. Venturi, TetR family member psrA directly binds the *Pseudomonas* rpoS and psrA promoters. *J Bacteriol* 184, 2324-2330 (2002).

6. T. Su *et al.*, The REC domain mediated dimerization is critical for FleQ from *Pseudomonasaeruginosa* to function as a c-di-GMP receptor and flagella gene regulator. *J Struct Biol* 192, 1-13 (2015).

7. C. L. Wysoczynski-Horita *et al.*, Mechanism of agonism and antagonism of the *Pseudomonasaeruginosa* quorum sensing regulator QscR with non-native ligands. *Mol Microbiol* 108, 240-257 (2018).

8. A. Jolma *et al.*, DNA-binding specificities of human transcription factors. *Cell* 152, 327-339 (2013).

9. A. Jolma *et al.*, DNA-dependent formation of transcription factor pairs alters their binding specificity. *Nature* 527, 384-388 (2015).

10. H. Huang *et al.*, An integrated genomic regulatory network of virulence-related transcriptional factors in *Pseudomonas* aeruginosa. *Nat Commun* 10, 2931 (2019).

11. C. E. Grant, T. L. Bailey, W. S. Noble, FIMO: scanning for occurrences of a given motif. *Bioinformatics* 27, 1017-1018 (2011).

12. T. Shimada, A. Ishihama, S. J. Busby, D. C. Grainger, The *Escherichia coli* RutR transcription factor binds at targets within genes as well as intergenic regions. *Nucleic Acids Res* 36, 3950-3955 (2008).

13. X. Yang *et al.*, A putative LysR-type transcriptional regulator inhibits biofilm synthesis in *Pseudomonas* aeruginosa. *Biofouling* 35, 541-550 (2019).

14. C. Baraquet, C. S. Harwood, Cyclic diguanosine monophosphate represses bacterial flagella synthesis by interacting with the Walker A motif of the enhancer-binding protein FleQ. *Proc Natl Acad Sci U S A* 110, 18478-18483 (2013).

15. J. W. Hickman, C. S. Harwood, Identification of FleQ from *Pseudomonasaeruginosa* as a c-di-GMP-responsive transcription factor. *Mol Microbiol* 69, 376-389 (2008).

16. C. Baraquet, K. Murakami, M. R. Parsek, C. S. Harwood, The FleQ protein from *Pseudomonasaeruginosa* functions as both a repressor and an activator to control gene expression from the pel operon promoter in response to c-di-GMP. *Nucleic Acids Res* 40, 7207-7218 (2012).

17. C. J. Jones *et al.*, ChIP-seq and RNA-Seq reveal an AmrZ-mediated mechanism for cyclic di-GMP synthesis and biofilm development by *Pseudomonas* aeruginosa. *PLoS Pathog* 10, e1003984 (2014).

18. C. J. Jones, C. R. Ryder, E. E. Mann, D. J. Wozniak, AmrZ modulates *Pseudomonasaeruginosa* biofilm architecture by directly repressing transcription of the psl operon.

*J Bacteriol* 195, 1637-1644 (2013).

19. M. D. Parkins, H. Ceri, D. G. Storey, *Pseudomonasaeruginosa* GacA, a factor in multihost virulence, is also essential for biofilm formation. *Mol Microbiol* 40, 12151226 (2001).

20. R. D. Monds, M. W. Silby, H. K. Mahanty, Expression of the Pho regulon negatively regulates biofilm formation by *Pseudomonas* aureofaciens PA147-2. *Mol Microbiol* 42, 415-426 (2001).

21. G. Rampioni, M. Schuster, E. P. Greenberg, E. Zennaro, L. Leoni, Contribution of the RsaL global regulator to *Pseudomonasaeruginosa* virulence and biofilm formation. *FEMS Microbiol Lett* 301, 210-217 (2009).

22. D. Badal, A. V. Jayarani, M. A. Kollaran, A. Kumar, V. Singh, *Pseudomonasaeruginosa* biofilm formation on endotracheal tubes requires multiple two-component systems. *J Med Microbiol* 69, 906-919 (2020).

23. N. S. Chand, A. E. Clatworthy, D. T. Hung, The two-component sensor KinB acts as a phosphatase to regulate *Pseudomonasaeruginosa* Virulence. *J Bacteriol* 194, 65376547 (2012).

24. S. Mukherjee, M. Jemielita, V. Stergioula, M. Tikhonov, B. L. Bassler, Photosensing and quorum sensing are integrated to control *Pseudomonasaeruginosa* collective behaviors. *PLoS Biol* 17, e3000579 (2019).

25. J. Zhao *et al.*, Structural and Molecular Mechanism of CdpR Involved in QuorumSensing and Bacterial Virulence in *Pseudomonasaeruginosa*. *PLoS Biol* 14, e1002449 (2016).

26. M. Guragain *et al.*, The *Pseudomonasaeruginosa* PAO1 Two-Component Regulator CarSR Regulates Calcium Homeostasis and Calcium-Induced Virulence Factor Production through Its Regulatory Targets CarO and CarP. *J Bacteriol* 198, 951-963 (2016).

27. S. Sarkisova, M. A. Patrauchan, D. Berglund, D. E. Nivens, M. J. Franklin, Calciuminduced virulence factors associated with the extracellular matrix of mucoid *Pseudomonasaeruginosa* biofilms. *J Bacteriol* 187, 4327-4337 (2005).

28. A. Finelli, C. V. Gallant, K. Jarvi, L. L. Burrows, Use of in-biofilm expression technology to identify genes involved in *Pseudomonasaeruginosa* biofilm development. *J Bacteriol* 185, 2700-2710 (2003).

29. J. Yan *et al.*, Transcription factor binding in human cells occurs in dense clusters formed around cohesin anchor sites. *Cell* 154, 801-813 (2013).

30. Y. Yin *et al.*, Impact of cytosine methylation on DNA binding specificities of human transcription factors. *Science* 356, (2017).

31. O. Wurtzel *et al.*, The single-nucleotide resolution transcriptome of *Pseudomonasaeruginosa* grown in body temperature. *PLoS Pathog* 8, e1002945 (2012).

32. K. Poole *et al.*, Expression of the multidrug resistance operon mexA-mexB-oprM in *Pseudomonasaeruginosa*: mexR encodes a regulator of operon expression. *Antimicrob Agents Chemother* 40, 2021-2028 (1996).

33. A. Kawalek, M. Modrzejewska, B. Zieniuk, A. A. Bartosik, G. Jagura-Burdzy, Interaction of ArmZ with the DNA-binding domain of MexZ induces expression of mexXY multidrug efflux pump genes and antimicrobial resistance in *Pseudomonasaeruginosa*. *Antimicrob Agents Chemother*, (2019).

34. U. H. Ha *et al.*, An in vivo inducible gene of *Pseudomonasaeruginosa* encodes an anti-ExsA to suppress the type III secretion system. *Mol Microbiol* 54, 307-320 (2004).

35. W. Wu, S. Jin, PtrB of *Pseudomonasaeruginosa* suppresses the type III secretion system under the stress of DNA damage. *J Bacteriol* 187, 6058-6068 (2005).

36. Y. Jin, H. Yang, M. Qiao, S. Jin, MexT regulates the type III secretion system through MexS and PtrC in *Pseudomonasaeruginosa*. *J Bacteriol* 193, 399-410 (2011).

37. A. Daddaoua, A. Corral-Lugo, J. L. Ramos, T. Krell, Identification of GntR as regulator of the glucose metabolism in *Pseudomonasaeruginosa*. *Environ Microbiol* 19, 3721-3733 (2017).

38. S. A. Chugani *et al.*, QscR, a modulator of quorum-sensing signal synthesis and virulence in *Pseudomonasaeruginosa*. *Proc Natl Acad Sci U S A* 98, 2752-2757 (2001).

39. J. Yan *et al.*, Systems-level analysis of NalD mutation, a recurrent driver of rapid drug resistance in acute *Pseudomonasaeruginosa* infection. *PLoS Comput Biol* 15, e1007562 (2019).

40. G. Dieppois, V. Ducret, O. Caille, K. Perron, The transcriptional regulator CzcR modulates antibiotic resistance and quorum sensing in *Pseudomonasaeruginosa*. *PLoS One* 7, e38148 (2012).

41. O. E. Petrova, K. Sauer, A novel signaling network essential for regulating *Pseudomonasaeruginosa* biofilm development. *PLoS Pathog* 5, e1000668 (2009).

42. I. Vallet *et al.*, Biofilm formation in *Pseudomonasaeruginosa*: fimbrial cup gene clusters are controlled by the transcriptional regulator MvaT. *J Bacteriol* 186, 28802890 (2004).

43. R. K. Ernst *et al.*, Specific lipopolysaccharide found in cystic fibrosis airway *Pseudomonasaeruginosa*. *Science* 286, 1561-1565 (1999).

44. E. L. Macfarlane, A. Kwasnicka, M. M. Ochs, R. E. Hancock, PhoP-PhoQ homologues in *Pseudomonasaeruginosa* regulate expression of the outer-membrane protein OprH and polymyxin B resistance. *Mol Microbiol* 34, 305-316 (1999).

45. M. M. Ramsey, M. Whiteley, *Pseudomonasaeruginosa* attachment and biofilm development in dynamic environments. *Mol Microbiol* 53, 1075-1087 (2004).

46. J. B. McPhee *et al.*, Contribution of the PhoP-PhoQ and PmrA-PmrB two-component regulatory systems to Mg^2+^-induced gene regulation in *Pseudomonasaeruginosa*. *J Bacteriol* 188, 3995-4006 (2006).

47. H. Liang, L. Li, Z. Dong, M. G. Surette, K. Duan, The YebC family protein PA0964 negatively regulates the *Pseudomonasaeruginosa* quinolone signal system and pyocyanin production. *J Bacteriol* 190, 6217-6227 (2008).

48. G. G. Nicastro, A. L. Boechat, C. M. Abe, G. H. Kaihami, R. L. Baldini,

*Pseudomonasaeruginosa* PA14 cupD transcription is activated by the RcsB response regulator, but repressed by its putative cognate sensor RcsC. *FEMS Microbiol Lett* 301, 115-123 (2009).

49. H. Mikkelsen, G. Ball, C. Giraud, A. Filloux, Expression of *Pseudomonasaeruginosa* CupD fimbrial genes is antagonistically controlled by RcsB and the EAL-containing PvrR response regulators. *PLoS One* 4, e6018 (2009).

50. S. Mukherjee, D. Moustafa, C. D. Smith, J. B. Goldberg, B. L. Bassler, The RhlR quorum-sensing receptor controls *Pseudomonasaeruginosa* pathogenesis and biofilm development independently of its canonical homoserine lactone autoinducer. *PLoS Pathog* 13, e1006504 (2017).

51. F. H. Damron, D. Qiu, H. D. Yu, The *Pseudomonasaeruginosa* sensor kinase KinB negatively controls alginate production through AlgW-dependent MucA proteolysis. *J Bacteriol* 191, 2285-2295 (2009).

52. W. Kong *et al.*, ChIP-seq reveals the global regulator AlgR mediating cyclic di-GMP synthesis in *Pseudomonasaeruginosa*. *Nucleic Acids Res* 43, 8268-8282 (2015).

53. V. Deretic, W. M. Konyecsni, A procaryotic regulatory factor with a histone H1-like carboxy-terminal domain: clonal variation of repeats within algP, a gene involved in regulation of mucoidy in *Pseudomonasaeruginosa*. *J Bacteriol* 172, 5544-5554 (1990).

54. D. Balasubramanian *et al.*, The regulatory repertoire of *Pseudomonasaeruginosa* AmpC ss-lactamase regulator AmpR includes virulence genes. *PLoS One* 7, e34067 (2012).

55. J. H. Hammond, E. F. Dolben, T. J. Smith, S. Bhuju, D. A. Hogan, Links between Anr and Quorum Sensing in *Pseudomonasaeruginosa* Biofilms. *J Bacteriol* 197, 2810-2820 (2015).

56. O. E. Petrova, K. Sauer, The novel two-component regulatory system BfiSR regulates biofilm development by controlling the small RNA rsmZ through CafA. *J Bacteriol* 192, 5275-5288 (2010).

57. O. E. Petrova, J. R. Schurr, M. J. Schurr, K. Sauer, The novel *Pseudomonasaeruginosa* two-component regulator BfmR controls bacteriophage-mediated lysis and DNA release during biofilm development through PhdA. *Mol Microbiol* 81, 767-783 (2011).

58. A. T. Yeung, M. Bains, R. E. Hancock, The sensor kinase CbrA is a global regulator that modulates metabolism, virulence, and antibiotic resistance in *Pseudomonasaeruginosa*. *J Bacteriol* 193, 918-931 (2011).

59. G. Wells, S. Palethorpe, E. C. Pesci, PsrA controls the synthesis of the *Pseudomonasaeruginosa* quinolone signal via repression of the FadE homolog, PA0506. *PLoS One* 12, e0189331 (2017).

60. I. Blus-Kadosh, A. Zilka, G. Yerushalmi, E. Banin, The effect of pstS and phoB on quorum sensing and swarming motility in *Pseudomonasaeruginosa*. *PLoS One* 8, e74444 (2013).

[Editors' note: further revisions were suggested prior to acceptance, as described below.]

[…] Although the manuscript has been improved, there are some remaining issues that need to be addressed, as outlined below:1. There are still concerns about the interpretations of RpoN binding data. It is well established that RpoN doesn't function as a dimer to facilitate transcription initiation. It binds RNAP core enzyme in a 1:1 molar ratio to form an RNAP holoenzyme. Second, the sequence motif itself is not a direct repeat of two half-sites. The 5' 'GG' and 3' 'GC' consensuses of the GGN(10)GC motif are representative sequences of the '-24' and '-12' elements of a typical sigma 54 (encoded by rpoN)-initiated promoter. The two elements are recognized by two separate domains of RpoN, both in the presence and absence of RNAP holoenzyme. This issue of data interpretation and representation needs to be addressed. It is also recommended that the authors use a different example of head to tail binding.

We deeply appreciate the reviewers for pointing out the potential issue of the data interpretation and representation of RpoN motif we included in previous submissions. We apologize for not carefully comparing the two similar motifs of RpoN identified in HT-SELEX. Putting the so-called “degenerated dimer” motif of RpoN in a reverse complementary order, we found that the motif was also composed of a “GG” and a “GC” half site. The difference lies on the fact that “GC” core sequence is on the 5’ position relative to the 3’ “GG” core sequence in the unknown motif. However, this previously uncharacterized motif, as explained by the reviewers, was not reproduced by any in vivo ChIP-seq data and may result from a possible topology in the in vitro biochemical condition (without interaction with other proteins) and therefore, we decided not to overstate the finding. Suggested by the reviewers, we replaced Figure 1C illustration of RpoN with another confident example of head-to-tail binding TF, PhoB, to avoid confusing readers. Agreed by the data from HT-SELEX and ChIP-seq by us (*1*) and by others (*2*), PhoB binds to DNA as a homodimer in a head-to-tail consecutive orientation, with a “GTCA(C/T)” monomer sequence preference spaced by a stretch of 6-bp ATrich nucleotides (see new Figure 1C). We have now added the text to the manuscript in Line 139-142.

2. The authors associate a TF with a pathway when a single predicted binding event occurs with any gene in that pathway. However, statistical analysis must be performed to determine the false discovery rate and whether their findings could occur via chance.

Thanks for the reviewers’ suggestion. We have now carried out enrichment analysis of TF binding sites located in promoter of genes involved in various functional pathways by the hypergeometric test. The *P* value was corrected by multiple test to obtain false discovery rate (method="fdr"). Then we associated a TF with a pathway when its binding sites were significantly enriched in promoters of genes in that pathway (FDR < 0.05). Finally, we counted the number of TFs significantly involved in each virulence pathway, and updated the numbers in the new Figure 2E. We have also modified the relevant description in the text, as shown in Lines 183-186. In addition, to highlight and better illustrate the major TFs involved in each pathway, we visualized the enrichment analysis results for each pathway using radar plots, as shown in "Figure 2—figure supplement 12" (cited in Lines 188, 343). In the plot, the radius represents the minus logarithm-transformed FDR, meaning that the longer the radius is, the more significantly the TF is associated with the pathway.

3. There are still points that the authors equate binding with function in the manuscript, please revise throughout to not make this overstatement.

Thanks a lot for pointing out this issue. We have now thoroughly gone through the text and revised the overstatements mixing the biological function with the putative TF binding. Please see changes highlighted in red font in the revised manuscript file.

References:

1. H. Huang *et al.*, An integrated genomic regulatory network of virulence-related transcriptional factors in *Pseudomonasaeruginosa*. *Nat Commun* 10, 2931 (2019).

2. P. Bielecki *et al.*, Cross talk between the response regulators PhoB and TctD allows for the integration of diverse environmental signals in *Pseudomonasaeruginosa*. *Nucleic Acids Res* 43, 6413-6425 (2015).